# Biocomposite thermoplastic polyurethanes containing evolved bacterial spores as living fillers to facilitate polymer disintegration

Han Sol Kim [1,8], Myung Hyun Noh [2,3,8], Evan M. White [4], Michael V. Kandefer [4], Austin F. Wright[4], Debika Datta [1], Hyun Gyu Lim[2], Ethan Smiggs[2], Jason J. Locklin [4], Md Arifur Rahman [5] ✉, Adam M. Feist [2,6] ✉ & Jonathan K. Pokorski [1,7] ✉

The field of hybrid engineered living materials seeks to pair living organisms with synthetic materials to generate biocomposite materials with augmented function since living systems can provide highly-programmable and complex behavior. Engineered living materials have typically been fabricated using techniques in benign aqueous environments, limiting their application. In this work, biocomposite fabrication is demonstrated in which spores from polymer-degrading bacteria are incorporated into a thermoplastic polyurethane using high-temperature melt extrusion. Bacteria are engineered using adaptive laboratory evolution to improve their heat tolerance to ensure nearly complete cell survivability during manufacturing at 135 °C. Furthermore, the overall tensile properties of spore-filled thermoplastic polyurethanes are substantially improved, resulting in a significant improvement in toughness. The biocomposites facilitate disintegration in compost in the absence of a microbe-rich environment. Finally, embedded spores demonstrate a rationally programmed function, expressing green fluorescent protein. This research provides a scalable method to fabricate advanced biocomposite materials in industrially-compatible processes.

Hybrid engineered living materials (ELMs) is a burgeoning field in which living and synthetic matter are combined to provide composite materials with augmented and complex functions, far beyond what a traditional polymeric material could accomplish alone. The promise of incorporating living matter into biocomposites has generated materials that are capable of responding to stimuli[1] (i.e. light, nutrients, inducers, etc), and consequently morphing their shapes[2,3] and/or

properties[4], which have been utilized for living biosensors[5], wearable bioelectronics[5,6], drug delivery systems[7], wound healing patches[8] and self-regenerating skin[9]. Introducing live cells into polymer composites as a sustainable and smart filler material has the potential to greatly improve both material properties and their ecological footprint. Live cells have ideal features as smart polymer additives such as self-replication, self-regulation, and programmable stimuli-responsiveness[10].

[1]Department of NanoEngineering, University of California San Diego, 9500 Gilman Dr., La Jolla, CA 92093, USA. [2]Department of Bioengineering, University of California San Diego, 9500 Gilman Dr., La Jolla, CA 92093, USA. [3]Research Center for Bio-based Chemistry, Korea Research Institute of Chemical Technology (KRICT), 406-30 Jongga-ro, Ulsan 44429, Republic of Korea. [4]New Materials Institute, University of Georgia, Athens, GA 30602, USA. [5]Thermoplastic Polyurethane Research, BASF Corporation, 1609 Biddle Ave., Wyandotte, MI 48192, USA. [6]The Novo Nordisk Foundation Center for Biosustainability, Technical University of Denmark, Building 220, Kemitorvet, 2800 Kgs, Lyngby, Denmark. [7]Institute for Materials Discovery and Design, University of California San Diego, 9500 Gilman Dr., La Jolla, CA 92093, USA. [8]These authors contributed equally: Han Sol Kim, Myung Hyun Noh. ✉e-mail: md-arifur.rahman@basf.com; afeist@ucsd.edu; jpokorski@ucsd.edu

Moreover, cells can be genetically programmed to autonomously synthesize both small and large molecules and can be further engineered to render other diverse functionalities[11,12]. Successfully harnessing living cells has limitless potential to develop polymer composites with enhanced properties such as improved mechanical performance and other performance characteristics, such as programmed/accelerated disintegration. However, live cells are fragile and require careful handling in terms of hydration, osmotic pressure, temperature and pH, compared to other non-living biological substances. Polymer processing into commercial parts typically requires heat, shear stress and/or solvents, all of which are detrimental to cell viability. To that end, incorporating live cells into polymers has only been demonstrated with limited types of polymers, which can be fabricated under mild conditions (i.e. low melting temperature, aqueous conditions, and low shear stress)[1,2,8,11]. To render cell-based biocomposite materials broadly useful, the poor stability of living cells must be resolved to use them in industrially-relevant fabrication processes.

Some bacteria have naturally evolved resistance to extreme conditions by forming spores[13]. Spores can preserve their viability against high temperatures, pressure, toxic chemicals (i.e. acids, bases, oxidants, and organic solvents), and radiation[14,15]. The robustness of spores is attributed to multiple factors that protect chromosomal DNA in the core, such as multi-layered protein coats, a peptidoglycan cortex, and a partially dehydrated core crowded with protective intracellular molecules[16,17]. Bacterial spores are metabolically dormant yet can survive for years. Even though spores have minimal metabolic activity, they are poised for germination into vegetative cells within minutes[18–21].

*Bacillus subtilis* are one of the most well-known spore-forming bacteria and have unique metabolic properties. They are ubiquitous in nature and FDA-approved generally recognized as safe (GRAS) substances[22,23]. Some *B. subtilis* strains have degradation activity toward polyester-based polymers[24,25]. Collectively, the high stability, safety, polyester degradation activity, and triggerable sporulation/germination of *B. subtilis* make them promising additives in developing biocomposite polymers with facilitated biodegradation. However, a significant challenge to employ *B. subtilis* spores in industrially-relevant polymer processing is their lack of heat tolerance. Spores of several *B. subtilis* strains are known to be resistant at ~100 °C for several minutes[26], but most industrial thermoplastic processing requires higher temperatures above 130 °C[27]. In fact, *B. subtilis* lost >90% of spore viability within 1 min at these temperatures[28,29].

Adaptive laboratory evolution (ALE) is an evolutionary engineering approach that has been effectively adopted in situations where the genetic causality to optimize a phenotypic property is not defined or intuitive[30–32]. By capitalizing on the natural occurrence of mutations during growth and division, coupled with growth-based selection, ALE effectively enhances desired phenotypes. This versatile approach finds wide applicability across various microorganisms that can be cultured in a laboratory setting. Notably, ALE has been effective for optimizing *Bacillus* species specifically in the past for enhancing tolerance towards inhibitors in hydrolyzed biomass[33], as well as stress tolerance to low pressures[34,35], and establishing a non-native pathway[36]. Thus, ALE is a well-suited strategy to augment the heat tolerance of *Bacillus* spores[37,38], thereby enabling their use as a functional living additive material.

We envisioned that the combination of evolutionarily engineered *B. subtilis* spores and polyester-based thermoplastic polyurethanes (TPUs) would have synergistic effects in both improving the mechanical properties and programming/facilitating the degradation of a spore-filled biocomposite TPU. The multi-layered proteins found in the exospore contain complex macromolecular structures which contribute to the glass transition temperature and, thus, the elastic moduli of bacterial spores[39]. Therefore, spores can simultaneously serve as particulate soft or rigid fillers to reinforce a TPU matrix, as well as biocatalysts for TPU disintegration under their dormant and germinated forms, respectively (Fig. 1). We demonstrated that heat-shock tolerized *B. subtilis* spores retained ~100% viability in TPUs after hot melt extrusion (HME). HME is the most widely used and industrially scalable technique in the polymer industry and forms the basis for most polymer manufacturing technology[40,41]. The resulting biocomposite TPU showed significantly improved tensile properties, when compared to neat TPU without spores or to non-evolved spore composites. Spores in the TPU matrix could be germinated by nutrients in compost and germinated *B. subtilis* cells improved the kinetics of end-of-life TPU disintegration in compost regardless of the microbial activity/diversity. Lastly, external biological function was successfully programmed into the biocomposite material by genetically engineering *B. subtilis* to express a model green fluorescent protein (GFP). Plasmids transformed into *B. subtilis* were retained in spores after HME, and fluorescence could be readily detected once the spores were germinated. Overall, this work presents a scalable method for the fabrication of biocomposite materials with improved mechanical properties and programmed biological functionalities.

## Results

### Adaptive laboratory evolution of *Bacillus* spores to increase heat-shock tolerance

To identify *Bacillus* strains with inherent TPU degradation and assimilation capabilities[42], we evaluated the growth of five common *Bacillus* strains (American Type Culture Collection; ATCC 23857, ATCC 6051, ATCC 7061, ATCC 21332, ATCC 6633) in minimal media supplemented with TPU powder (ground Elastollan® BCF45 gifted from BASF) as the sole carbon and energy source. Among the candidates, the ATCC 6633 strain exhibited robust growth and efficient assimilation of TPU powder, making it a suitable host strain for further study (Supplementary Fig. 1).

Previous research has shown that *Bacillus* spores[26] display robust heat-shock resistance but also exhibit differences in thermal tolerance across species. The ATCC 6633 strain has demonstrated significant heat tolerance in dry-heat conditions[43], however, we deemed this to be insufficient for melt processing and pursued an ALE experiment to further promote thermal stability of spores. We first characterized the initial heat-shock tolerance of spores by exposing them to boiling water (100 °C, i.e., wet-heat) for varying lengths of time (Fig. 2A) and the germination efficiency (colony forming units, CFU) of the treated spores compared to the untreated spores was assessed. Although the ATCC 6633 spores exhibited considerable heat tolerance in dry-heat conditions over 120 °C[26], exposure to wet-heat of 100 °C for even a few minutes resulted in a sharp decrease in viability. Only 24.7% of spores were able to germinate after 1 min of heat treatment, which further decreased to 3.3% after 3 min. TPU melt processing is performed at an even higher temperature, thus robust heat-shock tolerance is imperative for biocomposite fabrication.

To enhance the heat-shock tolerance of *B. subtilis* spores, ALE experiments were conducted (Fig. 2B, C). Spores were generated after 24 h of cell culture in Difco sporulation medium (DSM) and spore cultures were subjected to boiling water and then propagated into fresh DSM (Fig. 2B). The initial heating time was set to 3 min, which was enough to render >90% of spores dead. Subsequent heat tolerance passages were gradually increased to 30 min during the ALE experiments (Fig. 2C). Cell densities ($OD_{600}$) were measured at the start and after 24 h of culture to confirm sporulation. The ALE experiments were conducted for 40 cycles and were parallelly implemented with six independent lineages all from the same starting genotype to examine adaptive convergence[44]. All six ALE end-point lineages could continuously produce spores after 30 min of heat treatment (Fig. 2C), whereas the wild-type starting strain spore cultures could not show any detectable increase in cell density. This finding indicated the ALE

approach was effective in selecting for strains with enhanced heat-shock tolerance properties when in spore form.

## Identification of the genetic basis for enhanced heat-shock tolerance of *Bacillus* spores

To identify the causal mutations enabling the increased heat-shock tolerance of the ALE-derived strains, whole-genome sequencing was conducted for evolved populations and included two isolates from each of the final flasks of each ALE lineage ($n = 6$) (Supplementary Data 1). Two frequently mutated genes, *fusA* and *abrB*, were most prevalent across the parallel ALE lineages and were the focus of mutation causality assessment (Table 1). Mutations affecting these two genes made up 8 out of a total of 18 unique mutations (44.4%) identified from all samples (Supplementary Data 1). Overall, there was an average of 3 mutations per sample and a mode of 2 mutations for both population and clonal samples (lineages 4 and 6 had identical mutations indicating highly parallel evolution or potential cross-mixing). Notably, 5 different types of single nucleotide polymorphisms (SNPs) in *fusA*, encoding elongation factor G, were identified across all lineages and a *fusA* mutation was present in all samples sequenced (Supplementary Data 1). Additionally, 2 types of mutations in the upstream region of *abrB*, encoding transition state genes transcriptional regulator, were commonly mutated in 3 ALE experiments and their corresponding isolates (Table 1). Interestingly, one *abrB* mutation was a single SNP found 62 bp upstream of *abrB* (found in replicate lineage 5), and the other mutation identified was a combination of a 2 bp substitution (2 bp→TC) 52 bp upstream with a SNP 46 bp upstream (found in lineages 4 and 6). It is unclear if these mutation events happened at the same time during the experiment or if there was an order to their occurrence and this could be a subject of later analysis. Finally, there

were single occurrences of intergenic SNP mutations in *walH*, a two-component regulatory system regulator of *WalRK*, in lineage 2 and then a distinct SNP in *walK*, a cell wall metabolism sensor histidine kinase, in lineage 3. Both genes are in the same operon. Collectively, these convergent mutations suggested a strong influence of these genes on heat-shock tolerance[44].

To directly assess the effects of mutations selected on heat tolerance, we used several isolates with single or double mutations of *fusA* and *abrB* genes (Supplementary Data 1). The *fusA* gene encodes a translation elongation factor that is involved in protein synthesis, while the *abrB* gene encodes a transcription regulator that is involved in the control of various cellular processes, including sporulation[45,46] and stress responses[47]. Relative heat tolerances could be measured with germination efficiency after 10 min of heat treatment compared to wild-type ATCC 6633 strain (Fig. 2D). The isolate A3_F40_I1 (A3: ALE replicate #3, F40: Flask #40, I1: Isolate #1), with a single *fusA*^D560N mutation, the most frequently observed mutation, showed 3.0-fold increased germination efficiency compared to the wild-type strain. Meanwhile, isolate A4_F40_I1 had an additional *abrB* mutation along with *fusA*^D560N, through which the effect of the *abrB* mutation could be indirectly confirmed. The additional *abrB* mutation could bring a modest increase in heat-shock tolerance to 4.5-fold. Similarly, several unique *fusA* mutation effects could be deciphered along with *abrB* mutations; A5_F40_I2 isolate with *fusA*^V531A and *abrB* mutations showed a 2.5-fold increase, and A5_F40_I1 isolate with *fusA*^S309T and *abrB* mutations showed significantly increased heat-shock tolerance to 17.7-fold. Further research is required to fully understand the mechanisms underlying the observed increase in heat-shock resistance in the evolved *Bacillus* spores, but the causality analysis shows that mutations in these two genes resulted in increased heat-shock tolerance.

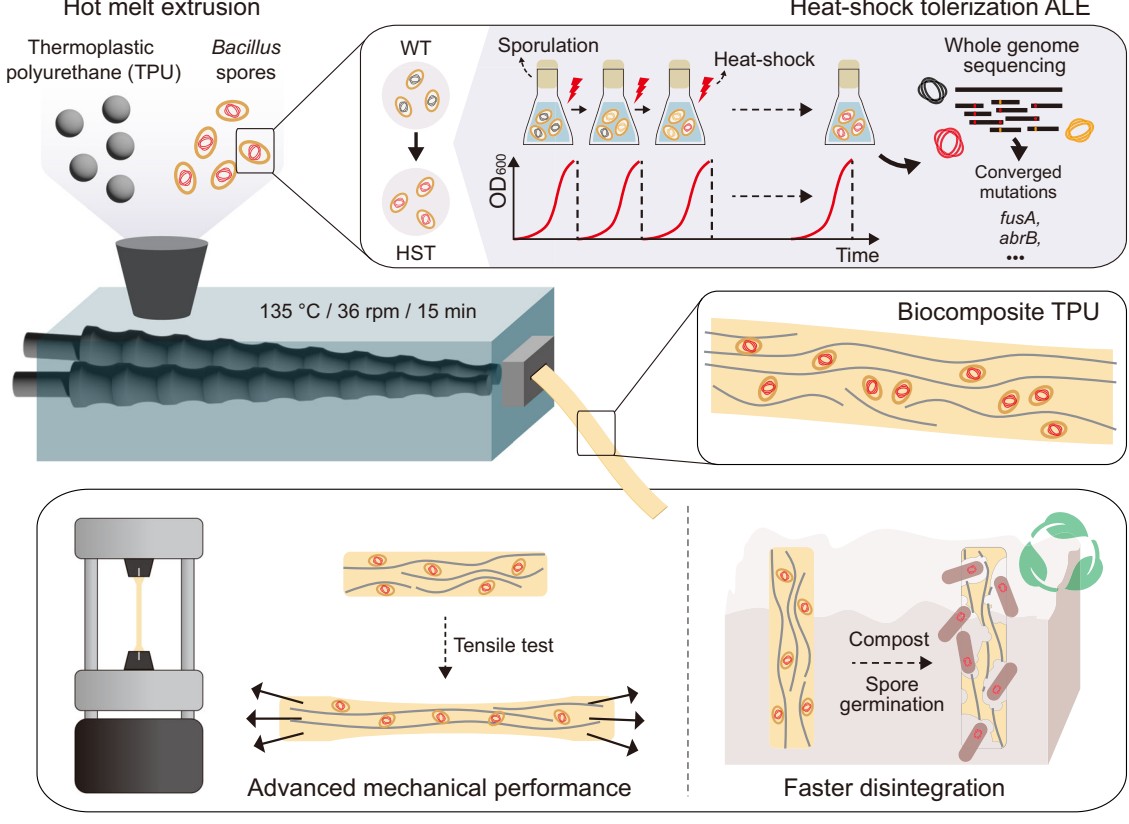

**Fig. 1 | Degradable biocomposite TPU.** Schematic representation of the fabrication of a biocomposite TPU filled with bacterial spores (WT wild-type ATCC 6633, HST heat-shock tolerized strain) capable of degrading TPUs. Heat-shock tolerance of spores was enhanced through adaptive laboratory evolution (ALE). The biocomposite TPU was fabricated via hot melt extrusion, wherein dormant spores acted as particulate fillers reinforcing the mechanical properties of TPU matrix. During the end-of-life testing, germinated bacterial cells enhanced disintegration of the TPU.

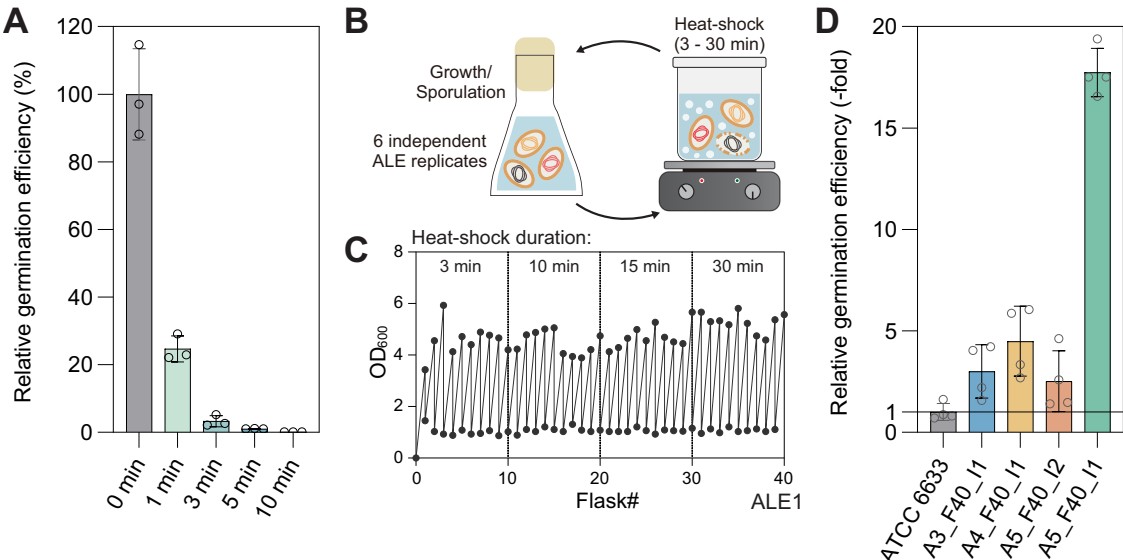

**Fig. 2 | Heat-shock tolerization of spore-forming bacteria via adaptive laboratory evolution.** **A** Characterization of initial heat-shock tolerance of *Bacillus subtilis* ATCC 6633 spores. Germination efficiency was measured after exposure to boiling water for different durations. Data are presented as mean values ± standard deviations from three independent experiments. **B** Conceptual design of an ALE experiment for improving heat-shock resistance of spores. **C** Representative growth profiles of one replicate of the ALE experiment **D** Characterization of enhanced heat-shock tolerance for ALE-derived clones after exposure to boiling water for 10 min. Data are presented as mean values ± standard deviations from four independent experiments. Source data are provided as a Source Data file.

## Biocomposite TPU fabrication by using heat-shock tolerized spores

Wild-type (WT) and heat-shock tolerized (HST, A5_F40_I1 strain) ATCC 6633 spores were incorporated into TPUs as bioactive additives during HME using a mini-twin screw extruder (TSE) (Fig. 1). Strain A5_F40_I1 was chosen for biocomposite fabrication as it showed the highest heat tolerance among the isolates screened and it only possessed 2 distinct mutations in the commonly mutated *fusA* and *abrB* genes (Supplementary Data 1). TPU pellets and lyophilized spores, up to 1 w/w%, were fed into the TSE and rigorously mixed at 135 °C at 36 rpm for 15 min, followed by extrusion through a slit die (slit size: 5.0 mm × 0.7 mm). As a result, biocomposite (BC) TPUs with WT or HST spores (BC TPU^WT or BC TPU^HST) were fabricated (Fig. 3A and Supplementary Fig. 2). Lyophilized spores possess an oblong shape with a ~500 nm diameter and ~1 μm length (Supplementary Fig. 3) and, thus, can serve as submicron particulate fillers if spores remain intact during the extrusion process. Both BC TPU^WT and BC TPU^HST displayed a light brown color following the incorporation of lyophilized spores (Fig. 3A). A uniform color distribution revealed proper mixing of spores and TPU melt during HME. It was spectrophotometrically determined that extracted spores from BC TPUs showed very consistent spectra to each other regardless of the sampling site from the extrudate, confirming uniform spore mixing (Supplementary Fig. 4).

To quantitatively determine the viability of spores post-HME, spores were extracted from the BC TPUs by dissolving the polymer component with *N,N*-dimethylformamide (DMF) using an established protocol[48]. The viability of extracted spores was assessed by using a CFU assay. WT spores displayed ~20% survivability after HME at 135 °C (Fig. 3B), whereas HST spores showed essentially full survivability (96–100%) regardless of spore loading (Fig. 3C). This finding indicated that the evolutionary engineering of *B. subtilis* spores by ALE markedly improved the heat tolerance of spores above the extrusion temperature of commercial thermoplastics, opening a new opportunity to fabricate biocomposite polymers. Full viability recovery after HME also suggested that the majority of the HST spores were highly resistant to the rigorous shear during HME (Supplementary Fig. 5). Notably, this enhanced heat tolerance was achieved without observation of phenotypic tradeoffs when compared to the parental strain, as both WT and HST strains exhibited similar cell growth profiles under multiple liquid media environments including TPU assimilation (Supplementary Fig. 6).

We visualized the 3-dimensional spore distribution in TPUs using micro-computed tomography (MicroCT) obtained using an X-ray microscope (XRM). XRM non-invasively revealed the successful incorporation of bacterial spores within the TPU matrices (Fig. 3D–F, Supplementary Fig. 7 and Supplementary Movies 1–3). Neat TPU has a high X-ray transmittance (~80% transmission), hence the intensity of XRM is primarily attributed to the enriched calcium ions (Ca²⁺) in the spore core[49]. Interestingly, XRM images of BC TPU^WT vs. BC TPU^HST exhibited stark differences. BC TPU^HST showed clear contrast between the TPU background and bright particulate spores, while the X-ray signal of BC TPU^WT was relatively low and scattered across the matrix with smaller diameters compared to that of BC TPU^HST. This finding is presumably due to structural damage of WT spores by heat and shear during HME. Ca²⁺ ions confined in the core of HST spores could be clearly detected, but XRM does not have sufficient resolution (700 nm spatial resolution and 70 nm voxel size) to detect dispersed Ca²⁺ ions from WT spore cores that were destroyed during extrusion. Well-defined particulates with high intensity in BC TPU^HST indicated a large population of HST spores retained their structure following HME.

## Table 1 | Commonly observed mutations from evolved isolates

| Region | Product | Unique mutations | Frequency (n = 18, final populations and clones) | ALE lineages |
|---|---|---|---|---|
| *fusA* | Elongation factor G | D560N | 10 | ALE1, ALE3, ALE4, ALE6 |
| | | V531A | 2 | ALE5 |
| | | S309T | 1 | ALE5 |
| | | E549K | 2 | ALE3 |
| | | D560G | 3 | ALE2 |
| *abrB* | Transition state genes transcriptional regulator | −46 T→A −51 CG→TC | 6 | ALE4, ALE6 |
| | | −62 G→A | 3 | ALE5 |

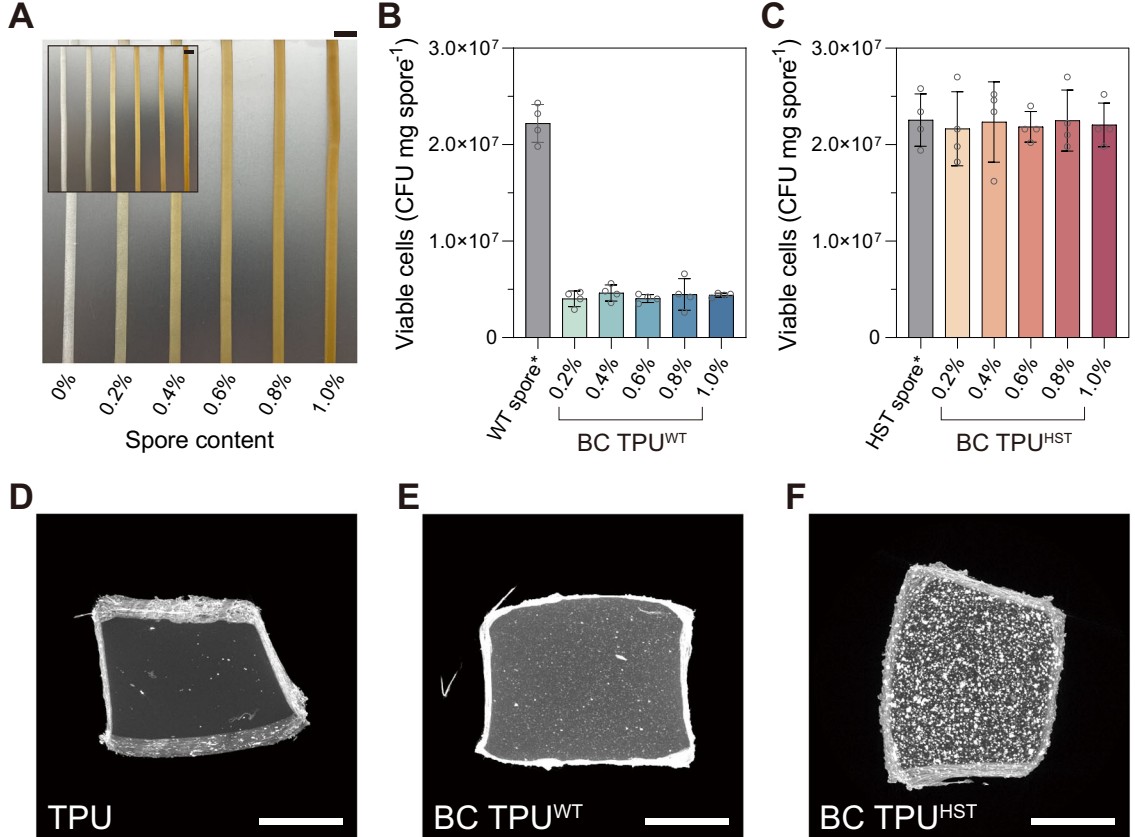

**Fig. 3 | Fabrication of spore-bearing biocomposite TPU. A** Photographs of BC TPU<sup>WT</sup> (inset) and BC TPU<sup>HST</sup> with 0, 0.2, 0.4, 0.6, 0.8, and 1.0 w/w% spore loadings, left to right (scale bars: 10 mm). The viability of WT (**B**) and HST (**C**) spores before and after HME. Spores were extracted from BC TPUs by dissolving the TPU component using DMF (no spore loss due to extraction assumed). Asterisk represents that the WT and HST spore controls followed the same solvent treatment procedures with the spores in BC TPUs. Data are presented as mean values ± standard deviations from four independent experiments. **D**–**F** MicroCT (XY projection) images of TPU and BC TPUs with 0.8 w/w% WT or HST spores (scale bars: 500 μm). Each XRM image was reconstructed by using 2401 projection images. XRM was presented without replication. Other XRM reconstruction images and movies are presented in Supplementary Fig. 7 and Supplementary Movies 1–3. Source data are provided as a Source Data file.

## Tensile properties of biocomposite TPUs

Spore incorporation generally had a positive effect on all mechanical properties measured for the BC TPUs. Tensile properties were evaluated based on four parameters; (i) toughness, (ii) elongation at break, (iii) ultimate tensile stress and (iv) Young's modulus (Fig. 4 and Supplementary Fig. 8) calculated from stress vs. strain curves (Supplementary Fig. 9). For example, BC TPU<sup>WT</sup> and BC TPU<sup>HST</sup> showed up to 25% and 37% enhanced toughness, respectively, compared to TPUs without spores (Fig. 4A, E). This finding affirmed that the spores behaved as reinforcing fillers, improving the tensile properties of the TPU matrix. Toughness improvement of BC TPU<sup>WT</sup> and BC TPU<sup>HST</sup> was most pronounced at spore loadings of 0.4 w/w%, 0.6 w/w%, and 0.8 w/w%, respectively. Additional loading above these critical concentrations led to a decreased toughness improvement, which was likely due to aggregation of spores within the matrix. High temperature and shear applied during the HME process, together with high spore concentration, has been found to induce the aggregation of spores in a TPU matrix[50]. Interestingly, the critical spore concentration of BC TPU<sup>HST</sup> (0.8 w/w%) was higher than BC TPU<sup>WT</sup> (0.4–0.6 w/w%), while the ultimate toughness improvement of BC TPU<sup>HST</sup> (37%) also outperformed that of BC TPU<sup>WT</sup> (25%) at their respective critical spore concentrations. These findings indicated that the heat tolerance of spores is important not only to retain their biological activities but also for better filler behaviors.

Toughness improvements of spore-filled TPUs are primarily explained by the increase in tensile strength (Fig. 4B, F) and matrix yielding (Fig. 4C, G) of composite materials. Tensile stress and

elongation at break of BC TPU<sup>HST</sup> was increased by up to 30% and 12%, respectively, by spore addition (Fig. 4F, G). Similarly, BC TPU<sup>WT</sup> showed up to 20% and 11% improvement in tensile stress and elongation at break, respectively, compared to TPU without spores (Fig. 4B, C). Increases in both the ultimate tensile stress and elongation at break of BC TPUs suggested a strong interfacial adhesion between the spores and TPU. Interaction between spores and TPU enhanced the energy dissipation in the composite systems based on stress transfer, as well as prevented the interfacial cavitation during the viscoelastic deformation of composite materials[51].

Interfacial adhesion between TPUs and spores was analyzed by an empirical model proposed by Pukánszky et al.[52]. The Pukánszky model was given for polymer composites containing quasi-spherical particles, which is appropriate for our composite system.

$$\sigma_c = \sigma_0(1 - \varphi)(1 + 2.5\varphi)^{-1} \exp(B\varphi) \tag{1}$$

$\sigma_c$ and $\sigma_0$ are tensile stresses of filled and neat polymers, respectively. $\psi$ is a volume fraction of filler in the matrix, which was calculated from the weight fraction by using the densities of the TPU (1.18 g cm<sup>−3</sup>) and *B. subtilis* spores[53] (1.52 g cm<sup>−3</sup>). $B$ is the load-bearing capacity of the filler, which depends on polymer/filler interfacial interaction[54]. The Pukánszky model was rearranged as:

$$\ln\left[\sigma_c(1 + 2.5\varphi)(1 - \varphi)^{-1}\right] = \ln\sigma_0 + B\varphi \tag{2}$$

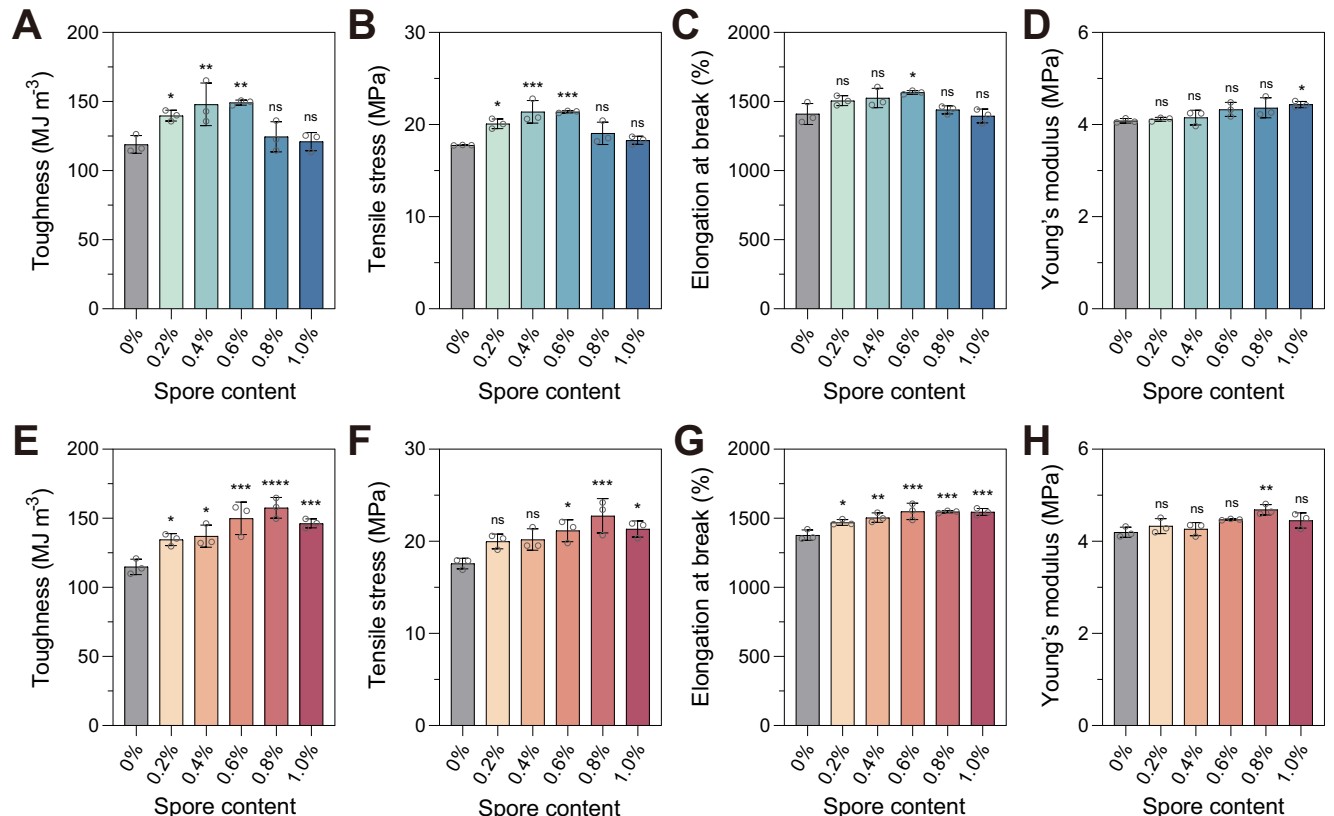

**Fig. 4 | Reinforcement effect of spores on the tensile properties of TPU.** Tensile properties of BC TPU with WT (**A**–**D**) or HST spores (**E**–**H**). Data are presented as mean values ± standard deviations from three independent experiments. One-way analysis of variance (ANOVA), followed by a post-hoc test with Dunnett's multiple comparisons (two-sided), was used for statistical comparison between TPU (control) and BC TPUs ($n = 3$ per group; ns: not significant; *$P < 0.05$; **$P < 0.01$; ***$P < 0.001$; ****$P < 0.0001$). Source data and detailed statistical analysis results are provided as a Source Data file.

The *B* value was obtained from the linear correlation between $\ln[\sigma_c (1 + 2.5\psi)(1-\psi)^{-1}]$ and $\psi$ (Supplementary Fig. 10). Data before the critical spore concentration was used for the plot[51]. Such a decrease in the y value of the linear form of the Pukánszky model above a certain filler content is oftentimes found in composite materials, which is likely due to the increased probability of matrix discontinuity and/or filler agglomeration. Both can behave as defect sites and initiate failure, negatively affecting the ultimate tensile strength[54]. The *B* values of BC TPU$^{WT}$ and BC TPU$^{HST}$ were estimated as 23.6 and 31.4, respectively (Supplementary Fig. 10). Such high *B* values clearly depicted a strong polymer/filler interfacial adhesion[55].

Even though both WT and HST spores served as reinforcing fillers for TPU with high *B* values, BC TPU$^{HST}$ showed a higher *B* value compared to BC TPU$^{WT}$, indicating stronger interfacial interaction between TPU and HST spores than that of TPU and WT spores. It can be explained by the (1) heat-shock tolerance of HST spores, which led to spores retaining their native structures (Fig. 3D–F) in TPU after HME and/or by (2) different interaction mechanisms between TPU and HST or WT spores. Water contact angle analysis revealed that the hydrophobicity of BC TPU$^{HST}$ corresponded well with the toughness improvement profile along with the spore content (Supplementary Fig. 11). The finding implied that among potential TPU/spore interactions, the hydrophobic interaction could be a critical driving force of improved tensile properties, particularly for BC TPU$^{HST}$. On the other hand, the hydrophobicity of BC TPU$^{WT}$ remained similar regardless of the spore loading, suggesting that either the surface hydrophobicity of WT spores was less than that of HST spores or the denatured spores did not build a strong hydrophobic interaction with the TPU. This hypothesis is consistent with the different *B* values between BC TPU$^{WT}$ (23.6) and BC TPU$^{HST}$ (31.4), which implied that different types of

interactions other than hydrophobic interactions may have contributed to the improved tensile properties of BC TPU$^{WT}$. Gel permeation chromatography (GPC) and attenuated total reflectance-Fourier transform infrared spectroscopy (ATR-FTIR) confirmed that the molecular weight and chemical composition of TPU did not change following HME, nor did the additional sample preparation steps for the water contact angle analysis impact polymer composition (Supplementary Figs. 12 and 13). The Young's modulus of BC TPUs was not significantly changed by the spore addition, suggesting that spores are soft filler materials and not affecting the stiffness of the TPU matrix (Fig. 4D, H).

**Facilitated disintegration of biocomposite TPU**

Facilitated disintegration of the BC TPUs was examined to determine the impact of spore incorporation, in general, and the impact that spore viability had post-HME. This analysis is relevant as spores embedded in the BC TPU can remain dormant until germination is triggered, typically by the sensing of nutrients[20,56], and reinforce the matrix as fillers throughout the lifespan of the TPU material. Then, at the end of the lifecycle of a BC TPU, spores can be germinated to facilitate TPU disintegration.

Approximately 80% of plastics are escaping recycling efforts and being accumulated in landfills or the natural environment[57]. Furthermore, polyurethanes (PU) are the sixth most produced plastic in world[58], but there is no governance for PU recycling. Even though PU waste can be potentially collected under category seven of resin identification code (for miscellaneous plastics other than PETE, HDPE, PVC, LDPE, PP, and PS), only 0.3% of plastics in this category are generally recycled in the United States[59]. In addition, the global infrastructure for industrial composting in a microbe-rich environment is

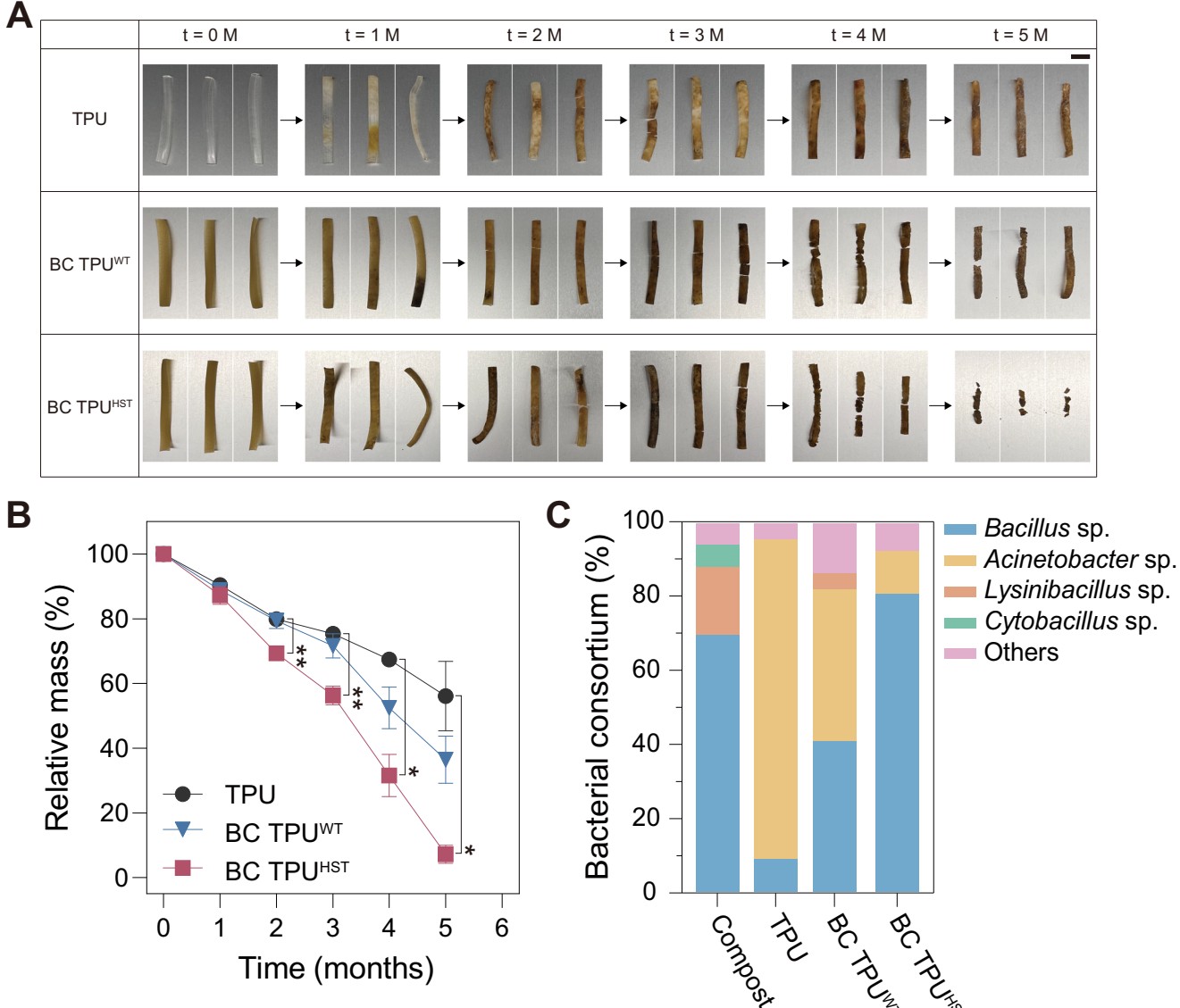

**Fig. 5 | Biodegradation test.** Photographs depicting the visual changes observed in the disintegration (**A**) and mass loss profile (**B**) of degraded TPU and BC TPUs during 5 months of incubation in autoclaved compost at 37 °C with 45–55% relative humidity (scale bar: 10 mm). Data are presented as mean values ± standard deviations from three independent experiments. Two-way ANOVA, followed by a post-hoc test with Dunnett's multiple comparisons (two-sided), was used for statistical comparison between TPU (control) and BC TPUs over time ($n = 3$ per data point; *$P < 0.05$; **$P < 0.01$). **C** Bacterial consortium analyzed from the autoclaved compost and TPU surfaces incubated in the autoclaved compost at $t = 4$ months. Source data and detailed statistical analysis results are provided as a Source Data file.

sparse and TPU waste would likely go to environments that are not enriched with TPU-degrading microorganisms at the end of its life-cycle. Thus, it is important to test the degradation of TPU in microbially active and, more importantly, less active environments under mild temperature conditions. We simulated these conditions by using microbially active compost and autoclaved compost, respectively, at 37 °C. A CFU assay determined that untreated compost was enriched with viable cells ($1.6 \times 10^6$ CFU mg compost$^{-1}$), whereby a minimal number of microorganisms survived (0.0002%) following autoclaving (Supplementary Table 1). After confirming the successful germination and growth of *B. sbutilis* spores using compost as a nutrient source (Supplementary Fig. 14), weight loss of BC TPUs in compost under controlled conditions of 37 °C and 45–55% relative humidity was assessed (Fig. 5 and Supplementary Fig. 15). BC TPU$^{HST}$ exhibited significantly faster disintegration (92.7% mass loss) compared to TPU (43.9% mass loss) and BC TPU$^{WT}$ (63.6% mass loss) in the autoclaved compost after 5 months (Fig. 5A, B). This suggested that both WT and

HST spores could be germinated post-HME and facilitate the disintegration of TPUs as living catalysts in an environment where few microbes were present when exposed to growth-supporting nutrients (Supplementary Table 1). Simultaneously, it indicated that the -100% survivability post-HME in HST spores was strongly beneficial for the TPU disintegration kinetics in such conditions. Furthermore, BC TPU$^{HST}$ in autoclaved compost showed comparable weight loss with that in unsterilized biologically active compost (Fig. 5B and Supplementary Fig. 15B). This finding indicated that the additional viability that HST provided can significantly aid in TPU fragmentation regardless of the inherent microbial activity in a given degradation environment. Disintegration of the TPU control without spores in autoclaved compost can be explained by the re-growth of degrader strains in compost during the incubation and/or abiotic hydrolysis of TPU.

To identify the microorganisms responsible for disintegration in compost, the surface of degraded TPUs was swabbed and analyzed using 16 S rRNA sequencing (Fig. 5C & Supplementary Fig. 15C). It

should be noted that the microbial analysis was focused on the bacterial community[60,61] considering their abundance in compost. In conjunction with the bacterial consortium analysis, supplementary sequencing analysis of *Bacillus* sp. was performed to evaluate the presence of the intentionally embedded strains. This process included the initial streaking from a swab onto an LB agar plate and efforts were made to minimize the variability in the initial bacterial community by ensuring ample incubation and employing random colony picking. Notably, bacterial strains could be characterized in autoclaved compost, while the number of colonies were notably reduced (Supplementary Table 1). *Bacillus* sp, *Lysinibacillus* sp., and *Cytobacillus* sp. were commonly observed as dominant as over 90% in both compost samples (Fig. 5C and Supplementary Fig. 15C). Interestingly, *Acinetobacter* sp. was dominant on the surface of pristine TPU by 80.0% and 86.4%, respectively in untreated and autoclaved conditions after 4 months, which were below the limit of detection in the initial compost. This suggested that *Acinetobacter* sp. likely exhibited significant TPU disintegration activity among the microbial consortium[42,62]. Some *Acinetobacter* were observed in BC TPUs, however a clear increase in the population of *Bacillus* sp. was confirmed as expected. BC TPU[HST] showed a higher proportion of *Bacillus* sp. (65.0% and 80.8% in untreated and autoclaved conditions, respectively) than BC TPU[WT] (40.0% and 40.9%). Additional sequencing analysis revealed that 61.5% and 85.7% of the *Bacillus* sp. from BC TPU[HST] in untreated and autoclaved compost, respectively, were the evolved strain incorporated during HME. These results indicated that the enhanced viability of HST spores within the BC TPU contributed to their ability to germinate and colonize on the surface and participated in disintegration. Overall, these findings confirmed that the HST spores can successfully germinate with biological activity in compost after fabrication and they indicated that the disintegration process is significantly accelerated in environments lacking sufficient quantity of degrader strains.

### Rationally programming biofunction into biocomposites through genetic engineering

Finally, the ability to rationally program the biofunction of ELMs is critically important to their overall ability to serve as smart materials. Thus, we genetically engineered the HST strain to demonstrate the introduction of a biofunction into biocomposite polymers. A plasmid harboring green fluorescent protein (GFP) was transformed into the HST strain and then incorporated into a BC TPU (BC TPU[HST/GFP]). Subsequently, BC TPU[HST/GFP] were incubated in either phosphate buffer saline (PBS), LB, or compost extract (CE) along with control samples (TPU and BC TPU[HST]). The composite materials were then imaged using confocal laser scanning microscopy (CLSM) analysis.

Imaging analysis confirmed the ability to rationally program a genetic function into embedded spores in polymer matrix post-melt extrusion and its activity in different environments. CLSM imaging qualitatively revealed the absence of green fluorescence signal from the TPU and BC TPU[HST] incubated in PBS or LB (Supplementary Fig. 16). There were weak fluorescence signals observed from BC TPU[HST/GFP] incubated in PBS (Fig. 6). This can be attributed to the residual GFP that was constitutively expressed during the bacteria cultivation and retained in or on spores during the spore purification process. In contrast, BC TPU[HST/GFP] incubated in LB and CE exhibited strong GFP signals across the TPU matrix (Fig. 6). This indicated that the spores within the BC TPU successfully germinated and self-replicated, utilizing the available nutrients, leading to GFP production. Furthermore, bright-field images obtained during CLSM analysis clearly showed the coincidence of spores and germinated cells on BC TPU[HST/GFP] (Fig. 6), providing evidence of spore germination within the composite material triggered by nutrients. These results indicated the plasmid could be retained and protected along with chromosome within HST spores during the HME process, and the desired function can be harnessed with genetic engineering.

## Discussion

Engineered living materials hold great promise to expand the usability and applications of polymeric materials. However, there are fundamental cellular physiological constraints in ELM fabrication and use. Primarily, a number of polymer/cell interactions need to be overcome and understood to generate viable materials. In this work, evolutionary engineering was critical to generate functional BC TPU materials that were able to disintegrate and maintain a genetically engineered heterologous expression system. Accordingly, the main findings from this work were that, (i.) the evolutionary engineering approach using a heat-shock tolerization ALE approach was effective in generating spores that could survive the HME process and the genetic basis of these mutations could be determined, (ii.) incorporation of heat-shock tolerized spores as living fillers into TPUs resulted in an overall improvement in the tensile properties of TPUs owing to the strong interfacial interactions between spores and polymer, and, (iii.) incorporation of spores into TPU to generate a living plastic material resulted in increased disintegration in compost. The results from each of these findings have implications for extension of the overall approach.

BC TPUs were fabricated by incorporating evolutionarily engineered *B. subtilis* spores into an HME process. ALE increased the heat-shock tolerance of *Bacillus* spores significantly without compromising overall fitness and strain usability (i.e., there were no apparent trade-offs). Furthermore, evolved cells were shown to effectively express a heterologous protein which expands opportunities for future applications. Sequencing and causality analysis revealed that highly converged mutations (a surprising 44% of all unique mutations identified across six replicates) in *fusA* and *abrB* enhanced heat-shock tolerance. These findings are consistent with previous studies that have shown mutations in genes encoding translation[63] and transcription factors[37,64,65] can confer increased stress tolerance in bacteria. Such specific mutations and targeted genetic regions would have been extremely difficult to predict and engineer using rational approaches. A unique aspect of this approach is that evolutionarily-engineered microorganisms derived from ALE are regarded as non-genetically modified organism (non-GMO) products given that ALE is facilitating and directing the natural evolution processes without artificial intervention[66]. It should be also noted that *B. subtilis* is not only beneficial for human health as probiotics[67-69], but also aids the growth of plants as biocontrol agents[70]. These advantages suggest the feasibility of fabrication and application of industrial BC TPUs utilizing the presented strategy.

The incorporation of heat-shock tolerized spores as living fillers into TPUs resulted in an overall increase in the tensile properties of TPUs owing to the strong interfacial interactions between spores and polymer matrix. This enhancement in tensile properties of TPU by spore addition offers a promising solution to overcome the tradeoff barrier between tensile stress and elongation at break in commercial TPUs (Supplementary Fig. 17). The evolved bacterial spores exhibited essentially full viability even after HME, clearly demonstrating a high level of biological functionality. The viability of spores following HME was critical for improvement of tensile properties and facilitating disintegration. Furthermore, we showcased the potential of genetic engineering by incorporating a GFP-expressing plasmid into the strains.

HME, used in this work, is an established technology for industrial polymer manufacturing and more than half of all plastic products are fabricated by HME[40,71]. The scale-up and scale-down of TSE-based HME is widely studied[72], and the throughput of BC TPU fabrication (15 g h⁻¹) by the benchtop TSE (Supplementary Fig. 18) can be potentially increased to larger scales. For example, our data showed that spores retain significant survivability (~88%) up to 144 rpm screw speed, which corresponds to up to 220 MPa shear stress (Supplementary Fig. 19). Increasing the screw speed, together with an increase in the screw and

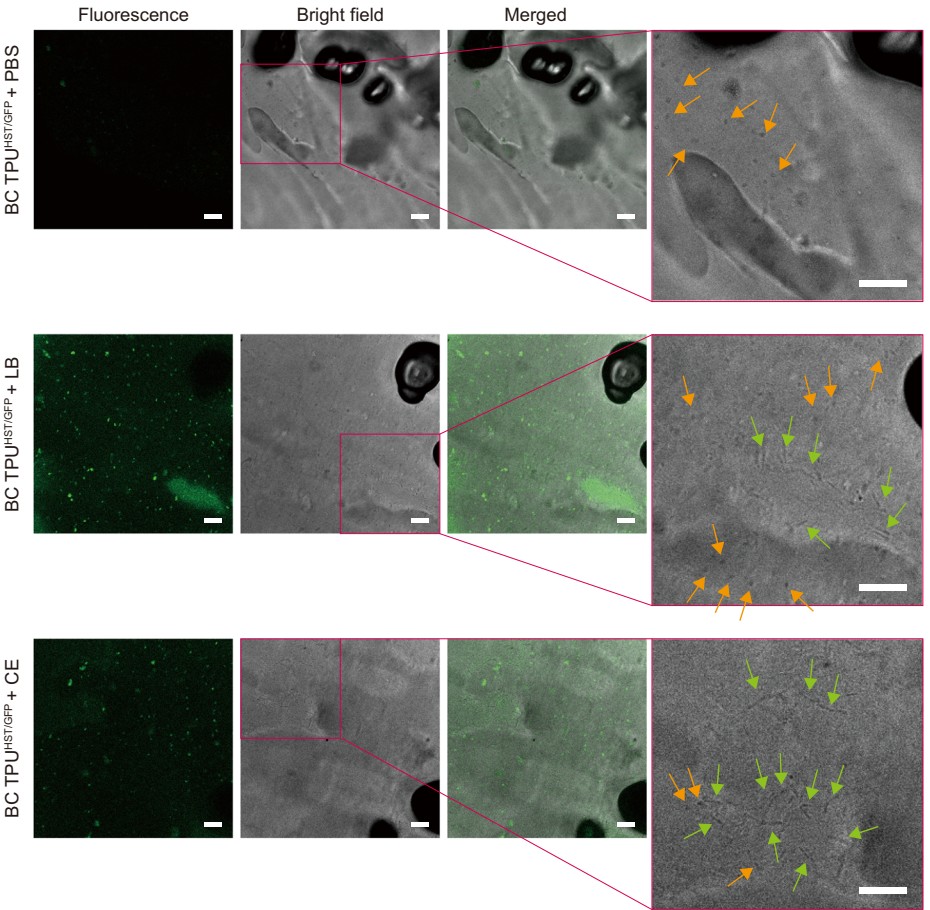

**Fig. 6 | Genetic engineering of spore-forming bacteria.** Fluorescence, bright-field, and merged images (left to right) of BC TPU$^{HST/GFP}$ incubated in PBS, LB, and CE obtained by CLSM. Rightmost panels are magnified bright-field images to identify the coincidence of rod-shaped vegetative cells with ~5 μm length (green arrows) and particulate spores with ~1 μm length (orange arrows). Scale bars: 10 μm. The experiments were repeated twice, and the representative images were presented. Other images are presented in Supplementary Fig. 16.

barrel size will achieve exponential upscaling. Under volumetric scaling, the throughput can be increased to the power of 3 with an increase in outer screw diameter (D)[73,74], yet the various relevant processing parameters are needed to be considered and optimized for the successful scale-up[75,76].

$$Q_i = Q_0 \left(\frac{D_i}{D_0}\right)^3 \frac{N_i}{N_0} \qquad (3)$$

where $Q$ is throughput, $D$ is outer diameter of screw and $N$ is screw rotation speed. Subscripts i and 0 represent two different scales. In addition, BC TPU fabrication is aligned with the workflow of the current practice of industrial TPU manufacturing, as lyophilized spores are a compatible dry additive.

Incorporation of evolved spores into the tested TPUs resulted in an increased disintegration and informs how the approach can promote fragmentation at the end of lifecycle in an environment void of robust microbial actively, as would often be the case in consumer polymer disposal. Future steps to engineer the cells for enhanced disintegration should facilitate disintegration for the targeted TPU and likely others[77]. It was demonstrated that BC TPUs in compost with robust microbial activity did not show an advantage in degradation (Supplementary Figs. 15 and 20), implying that this approach may not be necessary for such rich conditions. Nonetheless, the synergistic combination of genetic and evolutionary engineering can be further utilized to enhance TPU disintegration and introduce new biological functionalities to biocomposite materials.

Importantly, the practical use of TPUs relies on their mechanical properties. Thus, it is essential to maintain or improve these properties in developing biodegradable TPUs. The baseline TPU used in this work, BCF45, is a commercial grade TPU manufactured by BASF, which showed excellent tensile properties among TPU materials in the market (Supplementary Fig. 18). Spore incorporation not only facilitated the biodegradation but also improved the mechanical properties of BCF45 (i.e. tensile strength ~30%; elongation at break ~12%; toughness ~37%). There is a tradeoff barrier between the tensile strength vs. elongation at break of commercial TPUs (Supplementary Fig. 18), however the incorporation of HST spores moved the mechanical properties beyond this apparent barrier. State-of-the-art biodegradable TPUs are generally designed by blending the TPU with biodegradable polymers, such as cellulose, polylactic acid or polycaprolactone, or by optimizing the raw components for TPU synthesis to incorporate bio-based feedstocks[78–83]. However, tensile properties of these materials have not surpassed the tradeoff barrier[78–83]. Thus, BC TPUs developed in this work improve both tensile properties and biodegradation, which can potentially widen the application of TPUs.

This work focused on addressing the fate of TPUs in nature. Autoclaved compost simulated soil or landfill with marginal biological activity, while untreated compost fostered a microbially active condition for TPU biodegradation. Under both conditions, BC TPU$^{HST}$ achieved over 90% biodegradation at 37 °C in 5 months regardless of the microbial activity of the degradation substrate. This result depicted that nutrient and moisture content were sufficient for facilitating

biodegradation. In other words, spores embedded in the TPU matrix triggered and facilitated the biodegradation of TPU with minimal intervention. In comparison, most reports have tested the degradation of TPUs under specialized conditions[78–83], which often do not reflect the realistic fate of polymer waste.

In conclusion, the incorporation of bacterial spores presents exciting opportunities for the introduction of living cells as renewable polymer fillers in industrial processes. This innovative approach combines evolutionary and genetic engineering methodologies and shows potential for diverse applications in the advancement of biocomposite materials.

## Methods

### Bacterial cells, plasmids, and materials

*Bacillus* strains were obtained from the American Type Culture Collection (ATCC, Manassas, VA, USA). Q5® High-Fidelity DNA Polymerase, restriction enzymes, and NEBuilder® HiFi DNA Assembly Master Mix were purchased from New England Biolabs (NEB, Ipswich, MA, USA). Oligonucleotides for genetic manipulations were synthesized by Integrated DNA Technologies (IDT, Coralville, IA, USA). Soft-grade, polyester-based thermoplastic polyurethane pellets (Elastollan® BCF45) were gifted from BASF.

### Cell culture for bacterial growth and sporulation

Luria–Bertani (LB; $10 \, g \, L^{-1}$ tryptone, $5 \, g \, L^{-1}$ yeast extract, and $10 \, g \, L^{-1}$ NaCl) medium was utilized for routine cell culture (Fisher Scientific BP7923, Waltham, MA, USA). Difco sporulation medium (DSM) containing $8 \, g \, L^{-1}$ BD Difco™ nutrient broth (Fisher Scientific BD234000), $1 \, g \, L^{-1}$ KCl (Sigma–Aldrich P9333, St. Louis, MO, USA), and $0.12 \, g \, L^{-1}$ MgSO$_4$ (Sigma–Aldrich M2643) was used to sporulate *Bacillus* strains. Freshly prepared metal ion solutions were added into DSM before use to the final concentration of $1 \, mM$ CaCl$_2$ (Sigma–Aldrich C5670), $1 \, \mu M$ FeSO$_4$ (Sigma–Aldrich 215422), and $1 \, \mu M$ MnCl$_2$ (Sigma–Aldrich 221279). Minimal media containing $13 \, g \, L^{-1}$ 5X M9 minimal salts (Sigma–Aldrich M9956), $0.12 \, g \, L^{-1}$ MgSO$_4$, $0.11 \, g \, L^{-1}$ CaCl$_2$, $10 \, mL$ trace metal solution (Sigma–Aldrich 92949), and $10 \, \mu M$ FeSO$_4$ was used to investigate the TPU assimilation activity of *Bacillus* strains and the metabolic activity of ATCC 6633 WT and HST strains. Minimal media was also used to genetically engineer ATCC 6633 strain with GFP-containing plasmid. $10 \, g \, L^{-1}$ TPU powder, $4 \, g \, L^{-1}$ glucose (Sigma–Aldrich G7528) or $4 \, g \, L^{-1}$ glycerol (Sigma–Aldrich G5516) was supplemented to the minimal media as the carbon source depending on the experiment. All media were prepared in deionized water and autoclaved at 121 °C for 20 min.

LB agar plates were used for the isolation of strain and quantification of cell viability. LB agar plates were prepared by mixing LB and $15 \, g \, L^{-1}$ BD Bactor™ agar (Fisher Scientific BD214010). After autoclaving at 121 °C for 20 min, the mixture solution was transferred to petri dishes. Each petri dish contained 15 mL of the mixture solution. The mixture was gelated at room temperature. LB agar plates were flipped upside down and stored at 4 °C until use.

Glycerol stock was prepared by mixing cell cultured in LB medium under logarithmic phase of growth, indicated by an OD$_{600}$ of approximately 0.8, and 20% glycerol (Sigma–Aldrich G6279). The mixture was stored at −80 °C until use.

### Adaptive laboratory evolution (ALE)

Cell cultures for ALE were performed with a 30 mL cylindrical tube containing 15 mL of DSM at 37 °C and stirred at 1100 rpm. Seed cultures were prepared by inoculating individual colonies from LB agar plates for each ALE experiment. Sporulation could be achieved with cultivation for 24 h in DSM. Subsequently, the cultures were adjusted to a final concentration of OD$_{600}$ 1.0 and subjected to heat-shock treatment by immersing the tube in boiling water (100 °C). Following the heat-shock treatment, the cell cultures were centrifuged at

$2200 \times g$ for 10 min (5425, Eppendorf, Hamburg, Germany), and the resulting cell pellet was resuspended in fresh media for the subsequent passages, achieving a final concentration of OD$_{600}$ 0.3–0.4.

### Genome sequencing analysis

Genomic DNAs were prepared by using Mag-Bind® Bacterial DNA purification kit from Omega BioTek (Norcross, GA, USA) and converted into a sequencing library by using NEBNext® Ultra™ II DNA Library Prep Kit (Ipswich, MA, USA). Sequencing was performed by using Illumina Novaseq 6000 (San Diego, CA, USA) at UC San Diego IGM Genomics Center. Sequencing raw reads were processed with Breseq (version 0.35.4)[84] and a frequency cutoff ≥ 0.25 was applied for mutation analysis (Supplementary Data 1).

### Spore production and purification

Glycerol stock of *B. subtilis* was thawed and inoculated into 25 mL of LB at 1.25 v/v% to prepare seed culture. The seed culture was incubated at 37 °C at 200 rpm shaking overnight. 0.025 L of seed culture was then mixed with 2.475 L of DSM for the main culture and subsequent sporulation. The inoculated DSM was evenly distributed into six baffled flasks (4 L capacity) and incubated at 37 °C at 150 rpm shaking for two days in order to saturate and sporulate *B. subtilis*. *B. subtilis* spores were collected from the culture by the centrifugation at $12,000 \times g$ at 4 °C for 20 min (Avanti J-E, Beckman Coulter Life Science, Brea, CA, USA). Supernatant was removed and the spore pellets were resuspended in fresh 100 mM phosphate buffer saline (PBS) pH 7.4 (Thermo Fisher Scientific 10010023, Waltham, MA, USA). The recovered spores were washed by repeating the following steps for three times: (i) centrifugation at $2200 \times g$ at 25 °C for 10 min (5810 R, Eppendorf), (ii) removal of supernatant, and (iii) resuspension of spore pellet in fresh 100 mM PBS pH 7.4. The spores were then treated with $2.5 \, mg \, mL^{-1}$ lysozyme solution (Sigma–Aldrich 10837059001) at 37 °C at 150 rpm shaking for 1 h, followed by three times of washing with 100 mM PBS pH 7.4. The spores were further heat-treated at 65 °C for 1 h under static condition. After three times of washing by using deionized water, the spores were suspended in deionized water, frozen by using liquid nitrogen for 5 min, and thoroughly lyophilized for two days (Free-Zone 2.5, Labconco, Kansas City, MO, USA).

### Scanning electron microscopy (SEM)

SEM analysis of lyophilized spores was conducted by using FEI Apreo 2 SEM (Thermo Fisher Scientific). Spores were coated with gold for 60 s for 3 times. Images were obtained under high vacuum at 20 kV at 3.2 nA.

### Hot melt extrusion (HME)

Biocomposite TPUs were fabricated by using a TSE (HAAKE™ MiniCTW, Thermo Fisher Scientific) equipped with a slit die (slit size: 5.0 mm × 0.7 mm). ~5 g of TPU pellets were loaded to the TSE and cycled at 135 °C at 36 rpm for 5 min under cycle mode. Then, the lyophilized spore powder was introduced to the TSE at various loading ranging from 0 to 1.0 w/w%. The TPU-spore mixture was further processed at 135 °C at 36 rpm for 15 min under cycle mode and then extruded at 3 rpm under flush mode.

### UV-Vis spectrophotometry

Three 100 mg pieces of biocomposite TPU with 0.8 w/w% spore loading were collected from different sites in an extrudate. Each biocomposite TPU piece was diced into small pieces (~1 × 1 × 1 mm³), and dissolved in 10 mL *N,N*-dimethylformamide (DMF, Sigma–Aldrich 319937) at 40 °C under 150 rpm magnet stirring for 1 h. UV-Vis absorbance spectra of dissolved biocomposite TPUs and other controls were obtained at 260-700 nm wavelength by using a microplate reader (Synergy HT, BioTek, Winooski, VT, USA).

## Spore extraction from biocomposite TPU

100 mg of biocomposite TPU was diced into small pieces (~1 × 1 × 1 mm$^3$), and dissolved in 10 mL DMF at 40 °C under 150 rpm magnet stirring for 1 h. The solution was then centrifuged at 5000 g at 25 °C for 10 min. Supernatant was removed and the recovered spore pellet was resuspended in 10 mL of fresh DMF. The spore suspension was further soaked in DMF at room temperature at 50 rpm rocking for 10 min. After removing DMF by centrifugation at 5000 × g at 25 °C for 10 min, the spore was dispersed in 100 mM PBS pH 7.4.

## Cell viability test

The viability of spores was determined by colony forming unit (CFU) assay on LB agar plates. The viability could be accessed with 100 μL of serially diluted spore suspension in 100 mM PBS pH 7.4 spread onto LB plates. After 12 h incubation at 37 °C, the number of colonies appearing on LB plates was counted either manually or with image analyzing software (ImageJ). The concentration of viable cells in original spore suspension was calculated by considering the dilution factor.

## X-ray microscopy (XRM)

MicroCT images of TPU and biocomposite TPUs were obtained using Xradia 510 Versa (Zeiss, Oberkochen, Germany) operated at 80 kV and 7 mA. Samples were tailored into ~1.0 × 1.0 × 10 mm$^3$ bar shapes using a razor blade to be fitted into the sample holder of XRM. All the samples were analyzed without staining. Neat TPU and biocomposite TPUs were visualized at 4× magnification. X-ray exposure time was 1 s. 3D projection images and 3D volumes of microCT were constructed from tilt series with 2401 projections using XM3DViewer and XMReconstructor, respectively.

## Tensile testing

Biocomposite TPU was tailored into a dogbone shape by using a precision knife with a template for tensile testing (overall length: >38 mm; clamping area length: >10 mm; initial distance between grips: ~18 mm; length of narrow parallel-sided portion: 10 mm; width at ends: 5 mm; width at narrow portion: 2.4 mm; thickness: 0.7 mm). The dogbone specimen was loaded between two grips with serrated jaws. The specimen was stretched at 20 mm min$^{-1}$ extension rate until it fractured by using a universal testing machine (Instron 5982, Norwood, MA, USA) equipped with a 100 N load cell. Load (N) vs. extension (mm) curves obtained by the tensile testing were converted into stress (MPa) vs. strain (-) curves using the following equations.

$$\sigma = F/A \tag{4}$$

$$\epsilon = \Delta L / L_0 \tag{5}$$

where $\sigma$ is stress, $F$ is load, $A$ is cross-sectional area of the specimen, $\epsilon$ is strain, $\Delta L$ is displacement and $L_0$ is the initial length of the specimen.

Ultimate tensile stress and elongation at break of the specimen were obtained from the maximum stress value and strain value at the fracture moment, respectively. Toughness was calculated from the area under stress vs. strain curve based on the trapezoid rule. Young's modulus was calculated from the slope of stress vs. strain curve during the initial stretching of the specimen. Tensile properties of BC TPU$^{WT}$ or BC TPU$^{HST}$ were compared with their respective baseline TPUs, which were prepared on the same day of BC TPU$^{WT}$ or BC TPU$^{HST}$ fabrication via HME, respectively.

## Shear stress calculation

Shear stress during the HME was calculated from the following equations.

$$\tau = \frac{Tr}{J} \tag{6}$$

$$J = \frac{\pi}{2} r^4 \tag{7}$$

where $\tau$ is shear stress, $T$ is applied torque (obtained from MiniCTW), $r$ is radius of shaft, $J$ is polar moment of inertia.

Since conical screws were used for HME, radial distance ($r$) varied along with the length of screw. Thus, the maximum and minimum shear stresses were separately calculated from the minimum and maximum screw radii, respectively.

## Water contact angle

Prior to the water contact angle analysis, TPU samples were flattened by using a hot press. A small piece of TPU sample (~10.0 × 5.0 × 0.7 mm$^3$) was wrapped in aluminum foil and pressed between two flat heating blocks at 120 °C for 10 min through 6.3 MPa hydraulic pressure. Flattened TPU sample was transferred to a glass slide. The glass slide was loaded to a contact angle goniometer (Rame-Hart 500, Succasunna, NJ, USA). A droplet of water was dispensed and the contact angle was measured from the image taken by the contact angle goniometer.

## Gel permeation chromatography (GPC) analysis

The molecular weight of TPU before and after HME, as well as hot pressing, was analyzed by GPC (Shimadzu Prominence, Kyoto, JPN). TPU samples were dissolved in tetrahydrofuran (THF, Sigma–Aldrich 360589) at 10 mg mL$^{-1}$ final concentration at room temperature under 150 rpm magnetic stirring for 1 h. All TPU samples were filtered using 0.22 μm polytetrafluoroethylene (PTFE) syringe filters and introduced to a GPC equipped with Phenogel 5 μm 10$^4$ Å 300 × 7.80 mm GPC/SEC Column (Phenomenex, Torrance, CA, USA). Ultraviolet and refractive index detectors (Wyatt Technology, Goleta, CA, USA) were used to obtain chromatograms. THF (Sigma–Aldrich 34865) was used as an isocratic mobile phase at 1 mL min$^{-1}$ flow rate, and the oven temperature was equilibrated at room temperature. Polystyrene was used as a reference standard to calculate molecular weight.

## Attenuated total reflectance -Fourier transform infrared (ATR-FTIR) spectrometry

TPUs were analyzed by ATR-FTIR (NicoletTM iS50, Thermo Fisher Scientific) before and after HME, as well as hot pressing. Spectra was acquired at 4000–525 cm$^{-1}$ wavenumber with 4 cm$^{-1}$ spectral resolution at room temperature. 16 co-added scans were averaged. Potassium bromide (KBr) and deuterated-triglycine sulfate (DTGS) were used as beam splitter and detector, respectively.

## Compost sourcing

Raw compost aged 4–5 months was collected from two industrial composting facilities located in Athens, GA, specifically the University of Georgia and Athens-Clarke County Solid Waste Compost Facility. Landscaping and forest residues, food waste, and livestock manure are inputs for both facilities. The temperature of the composting pile at sampling depth (approximately 30–100 cm) was 46 ± 5 °C. The compost was particle sieved through a 4.76 mm screen. The sieved compost was mixed thoroughly with a resulting pH of 7.4 at the beginning of testing. See Supplementary Table 2 for compost details.

## Spore germination on compost gel

Compost was thoroughly dried at 80 °C under vacuum overnight. Dried compost was sieved by using a 35-standard mesh screen to yield fine powder. The sieved compost and agarose were suspended in deionized water, followed by autoclaving at 121 °C for 20 min. The final concentration of compost powder in suspension was varied from 0 to 100 g L$^{-1}$, while the concentration of agar was fixed at 15 g L$^{-1}$. After autoclaving, the liquid part of the mixture was gently taken by using an electrical pipette and transferred to petri dishes. Each petri dish contained 15 mL of compost extract-agarose mixture solution. The mixture was gelated at room temperature. Compost gel plates were flipped upside down and stored at 4 °C until use. Germination of spores on compost gels followed the same procedure with that on LB plates. In detail, 100 μL of serially diluted spore suspension in 100 mM PBS pH 7.4 was spread onto compost gel plates. After 12 h incubation at 37 °C, the number of colonies appearing on compost plates was counted manually. The concentration of viable cells in the original spore suspension was calculated by considering the dilution factor.

## Gravimetric analysis of TPU disintegration

Initial mass of each TPU piece used for the gravimetric disintegration test was 200 mg with ~48.0 × 5.0 × 0.7 mm$^3$ dimension. At least 30 pieces of each TPU or BC TPU sample were incubated in the untreated and autoclaved compost. Spore loading of BC TPU$^{WT}$ and BC TPU$^{HST}$ samples used for the disintegration test was 0.8 w/w%. Autoclaved compost was prepared by autoclaving compost at 121 °C for 20 min. 9 pieces of TPU samples were incubated in 500 g compost (by dry weight basis) in each 2 L plastic container, which was drilled with ¼ inch air holes. The plastic containers were then placed in an incubator at 37 °C at 45–55% relative humidity. The interior of the incubator was regularly disinfected using 70% ethanol after removing the plastic containers to prevent the (cross)contamination of autoclaved compost. 3 pieces of each TPU sample were collected every month. Large fragments of TPU were carefully collected from the compost, and the small fragments potentially remained in the compost was recovered after thoroughly drying compost, followed by sieving with a 35-standard mesh screen. The collected TPU samples were excessively washed with deionized water and dried at 80 °C under vacuum for overnight. The weight of degraded TPU samples was measured.

## Respirometry evaluation of TPU degradation

Respirometry evaluation was carried out using respirometry (ECHO, Slovenske Konjice, Slovenia) to evaluate the extent of mineralization of the TPU. (see Supplementary Table 2 for compost details). The evolution of $CO_2$ was quantified to determine the extent and rate to which sample carbon is consumed by microbes in the aerobic composting environment. Industrial composting conditions outlined in ASTM D5338 – 15/ISO 14855 maintain temperatures of 58 ± 2 °C for the duration of the testing; however, the conditions were modified to 42 ± 0.5 °C to study the metabolism of mesophilic microbes. Moreover, industrial composting temperatures often fall below 42 °C during the maturation phase. Respirometry was carried out using untreated compost only because there is no reliable method to suppress the contamination of autoclaved compost in respirometer under the current practice. 7 g of each TPU, BC TPU$^{WT}$ and BC TPU$^{HST}$ was cryo-milled and mixed with 125 g of compost (by dry weight basis). Bioreactors were purged continuously with 200 mL min$^{-1}$ air held at 42 ± 0.5 °C and stirred once weekly to avoid channeling and maintain moisture at 45–60%. Methane concentrations were measured and found to be negligible. The evolved $CO_2$ for each sample, $s$, was calculated daily by the difference in $CO_2$ production (mg day$^{-1}$) of each reactor, $r$, compared to the daily average $CO_2$ contribution from blank controls, $b$, as

described by the following equations:

$$s = r - b \qquad (8)$$

$$\text{Absolute Biodegradation}(\%) = \frac{s}{mc\left(\frac{44.01}{12.01}\right)} \qquad (9)$$

The sample mass, $m$, and the percent organic carbon content was used to determine the carbon content, $c$. The organic carbon and nitrogen contents of the polymers were determined with a Vario Elementar EL/max elemental analyzer using a TCD detector (Elementar Analysensysteme GmbH, Hanau, Germany) (Supplementary Table 3). The dimensionless value of 44.01/12.01 is used to account for the organic carbon in the $CO_2$ generated from each reactor. Relative biodegradation was calculated from absolute biodegradation of samples relative to that of cellulose, a positive control.

## Microbial consortium analysis in compost and composting experiments

To analyze the bacterial consortium, samples of compost or TPU obtained from the composting environment were collected. Sterilized swabs were used to gently swipe the samples onto LB agar plates. The plates were then incubated at 37 °C overnight. From each sample, at least 20 individual colonies were randomly selected. These colonies were subjected to characterization using 16 S rRNA sequencing (Eton Bioscience, San Diego, CA, USA/Macrogen, Seoul, South Korea) with primers (27 F - AGAGTTTGATCMTGGCTCAG, 1492 R - TACGGYTACCTTGTTACGACTT). The obtained sequences were compared to existing databases using BLAST (blast.ncbi.nlm.nih.gov) to identify the bacterial species. For the clones identified as *B. subtilis*, additional sequencing was performed using primers (fusA_F - ATGGCAAGAGAGTTCTCCTTAGACAAAACT, fusA_R - GATAATTTCTTCTGAAACGCTCTTCGGCAC) to ensure the embedded strains.

## Genetic engineering

To create GFP-expressing plasmid, the Cas9 gene cassette in pAW016-2[85] (Addgene, Watertown, MA, USA) was replaced with the GFP expression cassette using primers (GFP_F - TAGGAGGCAAAAATGGCTAGCAAGGGCGAGGAGCTGT, GFP_R - AGGTTTCGCGGCCGCTCACTTGTACAGCTCGT). The plasmid was introduced into the HST strain utilizing its natural competency[86]. Cells were cultured in minimal medium supplemented with 4 g L$^{-1}$ glucose at 37 °C and stirred at 1100 rpm until they reached the logarithmic phase of growth, indicated by an OD$_{600}$ of approximately 0.8, at which point plasmid DNA was directly added to the culture medium. The mixture was subsequently incubated for an additional 2 h to allow for DNA uptake, then cells were transferred to a selective medium, which contained 5 μg mL$^{-1}$ erythromycin (Sigma–Aldrich E5389) in LB. This antibiotic was used for selection purposes to ensure the growth of spores that successfully incorporated the plasmid.

## Compost extract (CE) preparation

CE was prepared by suspending dried compost powder in 100 mM PBS pH 7.4 at 100 g L$^{-1}$ final concentration. The compost suspension was autoclaved at 121 °C for 20 min, followed by the sedimentation under static condition for 30 min. The liquid part was carefully aliquoted for further experiment.

## Confocal laser scanning microscopy (CLSM)

Before CLSM analysis, TPU, BC TPU$^{HST}$, and BC TPU$^{HST/GFP}$ were diced into small pieces (~1 × 1 × 1 mm$^3$) and incubated in PBS, LB or CE at 37 °C under 150 rpm shaking overnight. 5 μg L$^{-1}$ erythromycin was added to the media for BC TPU$^{HST/GFP}$. The samples were then briefly rinsed with sterile PBS, then visualized using Leica SP8 confocal microscope

 

equipped with white light laser (Leica Microsystems) at 100x magnification. The channel for GFP detection was set for excitation at 484 nm and the emission was collected at 497–594 nm. Additional bright-field images were recorded. Multiple images from independent regions of each sample were recorded. CLSM images were post-processed using ImageJ. GFP signals of all images were enhanced by uniformly adjusting brightness/contrast threshold at 0–100. Minimum values of brightness/contrast threshold for bright-field images were fixed at 50 and the maximum values varied from 100 to 150 depending on the samples.

### Statistical analysis

Analysis of variance (ANOVA), followed by a post-hoc test with Dunnett's multiple comparisons (two-sided), was used for the statistical analysis. For the tensile properties, one-way ANOVA was carried out and the mean of each group was compared with a control group (TPU without spores). Two-way ANOVA was performed for the disintegration data with a factor of time and sample. GraphPad Prism 9.2 was used for the statistical analysis. $P < 0.05$ was considered statistically significant, and additional indicators of statistical significance are provided accordingly in the text or in individual figure legends. Detailed analysis results are included in the Source Data.

### Reporting summary

Further information on research design is available in the Nature Portfolio Reporting Summary linked to this article.

## Data availability

The raw resequencing data generated in this study have been deposited in the European Nucleotide Archive under accession code PRJNA981571 [https://www.ebi.ac.uk/ena/browser/view/PRJNA421359] and National Center for Biotechnology Information Sequence Read Archive under accession code PRJNA981571. The variant calling data is available in ALEdb v1.0[87] under project name BS6633_HSTALE [https://aledb.org/ale/project/166/]. Source data are provided with this paper. Any additional requests for information can be directed to, and will be fulfilled by, the corresponding authors. Source data are provided with this paper.

## Materials availability

ATCC 6633 HST strain is available through a material transfer availability.

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

## Acknowledgements

This work was primarily sponsored by funding from BOTTLE™ consortium (# DE-EE0009296, H.S.K., M.H.N., E.W., M.K., A.W., E.S., J.L., M.A.R., A.F. and J.P.) supported by the U.S. Department of Energy's (DOE's) Office of Energy Efficiency and Renewable Energy (EERE) and Advanced Manufacturing Office (AMO). This work was also in part sponsored by UC San Diego Materials Research Science and Engineering Center (UCSD MRSEC) (# DMR-2011924, D.D. and J.P.). H.G.L. and partially, A.F. was supported by the Joint BioEnergy Institute, U.S. Department of Energy, Office of Science, Biological and Environmental Research Program under Award Number DE-AC02-05CH11231. The authors acknowledge the use of facilities and instrumentation supported by UCSD MRSEC (# DMR-2011924), UCSD NanoEngineering Materials Research Center (NE-MRC), National Center for Microscopy and Imaging Research (NCMIR), and UCSD School of Medicine Microscopy Core (# NINDS P30NS047101). We also acknowledge the XRM expertise and operation consulting offered by Dr. Guy Perkins and Dr. Keun-Young (Christine) Kim from NCMIR. We would like to thank Paul R. Jensen and Alma Leticia Trinidad Javier of Scripps Institution of Oceanography, UCSD, for providing bacterial strains for screening during the project.

## Author contributions

H.S.K. and M.H.N. designed and performed experiments and analyzed the results. M.A.R. conceived the project and selected TPU material. E.W., A.W., M.K. and J.L. prepared compost, performed respirometry analyses, and consulted on gravimetric disintegration tests. D.D. performed CLSM. H.G.L. and E.S. supported ALE experiments. The overall project was supervised by A.F. and J.P. H.S.K., M.H.N., A.F. and J.P. wrote the manuscript with input from all authors. All authors contributed to the discussion of the results and the text.

## Competing interests

The authors declare no competing interests.
