## [Peer Review File · Nature Communications]

REVIEWER COMMENTS

Reviewer #1 (Remarks to the Author):

This article is of significant value because it outlines a strategy for melt extrusion of living organisms within a thermoplastic polymeric matrix. There is clear demonstration that the *Bacillus subtilis* spores, selected for through a process of adaptive revolutionary evolution (repeated exposure to increasing durations of boiling water), have increased heat shock tolerance and have survived 15 minutes of high temperature (135 degrees C) processing through a mini-twin screw extruder, in a cycling process that can be quite aggressive. There was also some effect on mechanical properties, with an increase in toughness, tensile strength and elongation at break. Biofunction was clearly retained, as evidenced both by the ability to biodegrade TPU in sterile compost as well as the ability for the genetically modified variants to retain fluorescence capability post processing.

There have been publications previously on incorporation of living organisms into polymer matrices. However, these have typically been prepared using low temperature (20 to 70 degrees C), low shear stress and/or water-based processes such as extrusion-spherulization, bioprinting (3D printing using UV promoted crosslinking to encapsulate organisms), microencapsulation, ram extrusion etc. Since this current work covers more commercially relevant extrusion conditions (135 degrees C), it is novel: although noting that this extrusion temperature is still lower than that of many higher melting commodity polymers.

The article is generally well written and logical, with the work supporting the conclusions and claims. However, there are some concerns that need to be addressed, as summarised below.

For this reason, I recommend that the paper be accepted, with revision.

Particular issues:

- 1) There has been no characterisation of the molecular weight pre- and post-processing. The extrusion protocol was quite aggressive, and with 15 minutes of cycling at 135 degrees C, there may have been thermal degradation of the polymer matrix. If this had occurred, this would potentially affect the biodegradation performance and the mechanical properties.

2) For contact angle determination there was an even further thermal processing stage (hot melt pressing). Again, the surface chemistry may have been modified through thermal degradation/oxidation. It would be good to do GPC analysis as well as ATR-FT-IR and/or NMR analysis of the materials before and post pressing to ensure that the changes observed are not artefacts - even though the control TPU went through the same stages, the presence of spores may have influenced the chemistry of degradation. I note that the UV-Vis spectra are less able to discriminate some of the characteristic degradation outcomes.

From a more general perspective, there were some other issues of note:

1) Throughout the text, the processing should be referred to as mini-twin screw extrusion; it is more like a model of the larger scale process. Likewise, the long (15 min) cycling process adopted is one that is feasible with this apparatus but far less likely in larger scale equipment and then only achieved through a complex screw profile - and that should be noted. Please provide the dimensions of the slit die. Further, polymers are not "incubated" in processing equipment, even those with living organisms incorporated. They are cycled, melt processed, or some other variant of such terms.

2) Need to define which standard method or variant of same was adopted for the tensile testing. For example, ASTM D882 is appropriate for sheets/films of <0.7 mm. Was the stretching rate consistent with standard? What load cell was used? How were dogbones cut? (Laser cutting, razor blade with a template - fresh blade every few cuts, other?)

3) In a couple of places it is mentioned that samples were collected "from different sites in an exudate". This is unclear. Should give the final dimensions of the ribbon produced post extrusion (there is usually shrinkage plus we don't know how long the extruded ribbon was) and then describe how sites were selected - random along ribbon? middle? edge?

4) Should specify the pneumatic pressure applied in hot pressing. I assume the ribbon pieces were used as-is and not placed in any mould or template?

5) Should specify source of compost at first use in the experimental. Note what depth compost was sampled from in the commercial facility.

6) How were samples prepared for gravimetric disintegration testing? What dimensions?

7) It is unclear to me why you would not do 16s rRNA testing directly on the swabs from the compost or TPU rather than going through a plating process first, given that so many native microbes are not readily culturable and will be missed in this process.

Other general comments:

1) There needs to be better justification as to why the biodegradation testing was done at a far lower temperature than typical industrial composting (we are not considering just the maturation phase). This more models home composting, which is fine if made clear. The fact that the *Bacillus* may not perform as well at the elevated temperatures of commercial composting may be worth noting.

2) The three different levels of magnification in Supp. Fig.2 do not add any additional information - suggest to remove at least one column of images.

3) Should be some discussion in the text that even with the data that was used for the Pukanszky model being already trimmed for data below the critical spore concentration, there was still a clear volume percent at which the samples no longer followed the linear relationship: there could be a number of explanations for this, but it should be commented on. It is also worth noting that the Pukanszky model was given for polymer composites containing quasi-spherical particles, which is appropriate for this composite application.

4) There is an odd plateau at around 25 days for cellulose biodegradation in Supplementary figure 11 B). It doesn't seem to match with the results in Supplementary figure 11 A). Is this right? Was there a reason for this?

Minor notes:

1) Please define the following at first use: GRAS, TSE, LB, PBS, SNP, HME

2) line 295: it is stress versus strain curves, not vice versa

3) lines 179, 258, 307, 391, 395, 416, 442: better to use indicate rather than indicated (the document uses both past and present tense inconsistently at present when indicating outcomes/conclusions). Likewise with "suggested" throughout.

4) Line 418: suggest: can successfully germinate with biological activity in compost after fabrication...

5) Line 504: screw geometry generally means screw profile and L/D, not just thickness - in this case you are just talking about barrel and screw diameter. The small-scale conical twin screw in this model system is not meant to be representative of a more complex twin screw profile with back flight, mixing, feeding and other zones. It should be recognised that the actual processing conditions at larger scale will be somewhat different.

6) line 559: introduced into the HST

7) line 593: 0.025 L

8) line 676: the polymer matrix dissolved, not incubated, in 10 mL DMF

9) line 679: spore suspension was further soaked at room temperature

10) line 694: followed by autoclaving

11) line 696: after autoclaving

12) line 707: the sieved compost was mixed thoroughly

Supplementary:

page 11: legend: with 45-55% relative humidity.

page 11: legend: spore loading. (B) Mass loss profile. Error bars indicate....

page 13: watch the significant figures in supplementary figure 11 D). In addition, line 156: specify that this is biodegradation relative to cellulose. Note: it is unclear if the shaded areas on the nonlinear curves represent the 95% confidence intervals for each sample or just shading between the three individual curves.

Line 162, starting with "As a result" is odd, and doesn't really follow from the previous text.

Supplementary figure 13 is not really clear. There are a lot of samples represented by different shapes (triangles, circles, diamonds etc.) without any definition as to what these symbols represent - we only know what the colour family indicates. Were other commercially available TPUs outside the Elastollan range included? If so, where did the commercial data come from?

Finally, in supplementary table 2, the solids content for the compost for respirometry seems low. Was this a wetter compost? If so, why?

Reviewer #2 (Remarks to the Author):

In this manuscript, Kim et al. generated a new kind of engineered living material (ELM), which was composed of evolved *Bacillus subtilis* spores and thermoplastic polyurethane (TPU). This biocomposite could be characterized and functionalized in different ways. First, the authors carried out adaptive laboratory evolution of *B. subtilis* ATCC 6633 spores to successfully improve the property of heat-shock tolerance (up to 135°C during hot melt extrusion). Next, the authors found two of the commonly mutated genes (*fusA* and *abrB*) that would enhance the heat-shock tolerance. Then, the fabrication of biocomposite by using TPU and evolved spores (spores of evolved A5_F40_I1 strain) was successfully performed and the tensile properties were evaluated in four different ways. Finally, such biocomposite TPU could be utilized to facilitate disintegration and genetically express GFP within cells. In conclusion, this manuscript exhibited a composite material consisted of living spores and nonliving TPU, with compelling mechanical properties and expanded biological applications.

Overall, the experiments in this work are well designed and performed, the data are solid and convincing, and the manuscript is well-written and clearly presented. Specially, the newly produced ELM by *B. subtilis* spores were embedded into polymers, rather than biofilms (Huang et al. *Nature Chemical Biology*, 15, 34-41, 2019; Zhang et al. *Materials Today*, 28, 40-48, 2019), agarose (González et al. *Nature Chemical Biology*, 16, 126-133, 2020), or biomineralized scaffolds (Kang et al. *Nature Communications*, 12, 7133, 2021), which provided another approach to construct *B. subtilis* spores-based ELMs. Such design may become a demo and provide inspirations for the further researches in this field. Taken

together, I highly recommend this manuscript for publication in Nature Communications after addressing the following issues.

Major concerns:

1. Page 4, line 75: The statement of “Despite this potential, live cells have rarely been exploited as polymer additives in practice due to their fragility.” is not appropriate. In recent years, there were many polymer-based ELMs that have been reported (e.g., bacteria in calcium alginate gel: Chen et al. Nature Chemical Biology, 18, 289-294, 2022; Peng et al. Advanced Materials, 2305583, 2023). I suggest the authors to revise this sentence.
2. Page 4, line 82: The statement of “and in most studies it is unclear if the cells maintained viability.” is not appropriate. As far as I know, most of published papers about ELMs contain the experimental results of cell viability (e.g., Tang et al. Nature Chemical Biology, 17, 724-731, 2021). I suggest the authors to revise this sentence.
3. Page 5, line 94: In my opinion, the statement here of probiotic properties on *Bacillus subtilis* is not suitable, because this manuscript lacked the utilization of its probiotic properties. I recommended the authors to replace the properties of “probiotic” with “GRAS (page 25, line 497)” here, and move the statements of probiotic properties to the Discussion section.
4. In Figure 3A, the photographs lacked scale bars. I suggest the authors to revise and add scale bars because it is important to indicate the lengths and diameters of the BC TPU fibers.
5. In the body text of this manuscript, the “Supplementary Fig. 11” was lacked between “Supplementary Fig. 10” and “Supplementary Fig. 12”. I suggest the authors to rearrange the order of Supplementary Figures in the revised manuscript files.

Minor concerns:

1. In Fig. 1, the “Hot-melt Extrusion” included the symbol of “-”, however, the other statements in this manuscript were the same as “hot melt extrusion”. Please unify the format.
2. In Supplementary Information, line 2 of “Table of Contents”, there needs a space between “ATCC” and “6633”.
3. Page 11, line 222: When the statements of “A3_F40_I1” was firstly presented in this manuscript, it needs a detailed explanation, like “A3: ALE lineage 3; F40: Flask 40; I1: Isolate 1”. Such explanation will help readers to clearly understand the meanings.
4. Page 12, line 243: The statement of “twin-screw extruder” included the symbol of “-”, however, the other statements in this manuscript were the same as “twin screw extruder”. Please unify the format.
5. Page 13, line 268: The typo of “using a X-ray microscope” should be corrected as “using an X-ray microscope”.

6. Page 16, line 320 and 327: Please carefully correct the authors' names as "Pukánszky", rather than "Pukanszky".
7. Page 24, line 464: The statement of "heat shock" lacked the symbol of "-", however, the other statements in this manuscript were the same as "heat-shock", including a symbol of "-". Please unify the format.
8. Page 25, line 483: The abbreviation of "non-GMO" should be clarified here for readers, may be "non-genetically modified organism".
9. Page 25, line 497: The abbreviation of "GRAS" should be clarified here for readers, may be "Generally Recognized as Safe".
10. Page 25, line 500: The abbreviation of "TSE" should be clarified here (Twin Screw Extruder?), or move to the above of this manuscript when the full name was firstly presented.
11. Page 27, line 533: The typo of "have not surpassed the trade-off-barrier" should be corrected as "have not surpassed the trade-off-barrier".

Reviewer #3 (Remarks to the Author):

The manuscript by Han Sol Kim et al. presents a novel method for fabricating Bacillus spore-filled thermoplastic polyurethane (TPU) biocomposite. Temperature resistance of Bacillus subtilis spores was enhanced through Adaptive Laboratory Evolution (ALE) to fabricate novel ELMs via hot melt extrusion and to reinforce the TPU matrix's mechanical properties.

Developing a novel strategy to incorporate living cells into thermoplastic material could allow the utilization of ELMs for various applications and advance the additive manufacturing of ELMs.

Although the topic of this study is important, this manuscript has overinterpreted the presented data and possesses limited novelty. The authors have provided a lot of data from tedious experiments, but unfortunately, it does not clearly support the hypothesis and does not make this manuscript better.

Due to the many concerns, I have stated below, I would not recommend the publication of this manuscript in Nature Communication.

Major concerns:

1. Tensile properties of biocomposite TPUs.

First, the particular tensile properties of TPU WT and HST should be incorporated together on one graph. It is difficult to compare them if they are not on the same graph.

Second, in line 306, the authors claim that the results (difference between WT and HST) are remarkable. I could not call the 12% difference in toughness remarkable.

From the graphs, it looks like WT TPU has lower spores (0.4%), resulting in higher toughness than 0.4% of HST TPU. You need to add double the amount of HST TPU spores (0.8%) to achieve slightly better toughness. This needs to be addressed in the manuscript, and claiming that the mechanical properties of HST TPU are remarkable is an overstatement.

Third, the HST TPU shows only a 10% increase in tensile test and a 1% increase in elongation break. These results also are not exciting to claim such remarkable mechanical properties improvement.

2. Facilitated disintegration of biocomposite.

Here, the authors wrote that at the end of the life cycle of a BS TPU, spores can be germinated to facilitate TPU disintegration.

First, the authors did not provide what % of spores integrated into TPU were carried in described degradation experiments.

Second, the difference in mass loss between WT and HST is 29.1% after 5 months in autoclaved compost. Why there is so little difference?? This difference should be significantly larger if WT had only 20% viable cells and HST ~100%. I assume the HST variant has negatively altered metabolic activity, so they cannot degrade TPU efficiently.

Third, in Supplementary Fig 10B. authors show that TPU alone, TPU-WT, and TPU-HST show the same level of mass loss after 5 months in non-autoclaved compost. No difference. It means that even TPU alone, wherein spores are absent, degrades to the same extent. Thus, the statement that engineered material facilitated biocomposite's disintegration seems flawed. The results do not support the hypothesis.

3. In line 117 – The authors say they are “programming/facilitating degradation of a spore-filled biocomposite TPU”. This claim is overstated. The degradation is only facilitated, *Bacillus* has natural degradation activity against polyester. In this work, the degradability of TPU is not programmed or engineered but rather evolved.

4. In line 135 – “Overall, this work presents a scalable method for fabricating biocomposite materials with improved mechanical properties and programmed biological functionalities.”

There are no programmed biological functionalities. GFP fluorescence in *Bacillus* pores has already been shown in González, L.M., Mukhitov, N. & Voigt, C.A. Resilient living materials built by printing bacterial spores. *Nat Chem Biol* 16, 126–133 (2020)

5. The authors use ALE experiments to increase the heat tolerance of *B. subtilis* spores. The experiment was successful, giving survivability of about 96-100% compared to 20% WT spores after hot melt

extrusion at 135 C. Next, the authors focused on identifying the mutations that lead to heat tolerance. I find this study interesting. However, the authors missed the most important point that it is the most significant to develop novel ELMs.

To develop functional ELMs, living bacteria should be viable and metabolically active to perform programmable functions. The ALE, as the evolutionary method, can influence metabolic activity significantly. The authors focused only on variants with growth rates and viability similar to WT. Still, they did not test if they alter metabolic activity, which is very easy to check via various assays. I find it a large overlook that can lead to developing ELMs that could not be functional even if bacteria are alive.

6. Line 264 – The authors wrote that WT and HST strains exhibited similar specific growth rates, yields, and cell viability. No data is provided. It should be provided as a supplementary figure.

7. Figure 3D-F – The TPU and BC TPU images are of odd, slopy shapes. The authors did not care to fabricate a composite with a uniform shape. I personally believe that better samples and images are suitable for publication in journals like Nature Communication.

8. Programming biofunction.

Authors genetically engineer HST strain to produce GFP. It has been done previously at González, L.M., Mukhitov, N. & Voigt, C.A. Resilient living materials built by printing bacterial spores. *Nat Chem Biol* 16, 126–133 (2020). The authors provide fluorescence images without fluorescence quantification. It is difficult to spot differences between biocomposite in LB and compost. The material's functionality is shown only at the end of ELMs' life after adding it to compost. The point of developing ELMs is to show functional material that can perform, not perform only during degradation.

Minor concerns:

1. In the introduction, the authors claim that ELMs are composed of living cells combined with composite materials. It is not quite correct. Authors describe hybrid ELMs. Besides hybrid ELMs, “autonomous” ELMs are made from engineered cells to produce functional material that embeds living cells. Autonomous ELMs started and pioneered this field. The authors do not mention it and do not cite any relevant literature.

2. Supplementary Figure 2. The SEM images of spores show lyophilized images of spores. The manuscript suggests that they are images of spores after fabrication with TPU. It is misleading. It could be useful to show SEM images of spores incorporated with TPU and a cross-section of this hybrid material.

3. Figure 3B, and C. should show the viability of WT and HST spores after HME. Graphs show the highest viability of cells with 0% of spore content. I believe it is a mistake, and this is a control - spore viability before HME treatment.

Please see our point by point response to the reviewers' comments. Comments from the reviewers are in regular font, our response is in blue, and text in red indicates changes made to the text or additional/revised figures. Likewise, changes in the text of the final manuscript are also indicated in red with highlighting.

Reviewer(s)' Comments to Author:

Reviewer #1 (Remarks to the Author):

This article is of significant value because it outlines a strategy for melt extrusion of living organisms within a thermoplastic polymeric matrix. There is clear demonstration that the *Bacillus subtilis* spores, selected for through a process of adaptive revolutionary evolution (repeated exposure to increasing durations of boiling water), have increased heat shock tolerance and have survived 15 minutes of high temperature (135 degrees C) processing through a mini-twin screw extruder, in a cycling process that can be quite aggressive. There was also some effect on mechanical properties, with an increase in toughness, tensile strength and elongaion at break. Biofunction was clearly retained, as evidenced both by the ability to biodegrade TPU in sterile compost as well as the ability for the genetically modified variants to retain fluorescence capability post processing.

There have been publications previously on incorporation of living organsims into polymer matrices. However, these have typically been prepared using low temperature (20 to 70 degrees C), low shear stress and/or water-based processes such as extrusion-speronization, bioprinting (3D printing using UV promoted crosslinking to encapsulate organisms), microencapsulation, ram extrusion etc. Since this current work covers more commercially relevant extrusion conditions (135 degrees C), it is novel: although noting that this extrusion temperature is still lower than that of many higher melting commodity polymers.

The article is generally well written and logical, with the work supporting the conclusions and claims. However, there are some concerns that need to be addressed, as summarised below.

For this reason, I recommend that the paper be accepted, with revision.

Particular issues:

1) There has been no characterisation of the molecular weight pre- and post-processing. The extrusion protocol was quite aggressive, and with 15 minutes of cycling at 135 degrees C, there may have been thermal degradation of the polymer matrix. If this had ocured, this would potentially affect the biodegradation performance and the mechanical properties.

Response: Thank you for that insightful comment. In response, we have performed additional experiments to determine chemical and physical properties of the TPU. We have performed GPC analysis of the TPU before and after hot melt extrusion, as well as hot pressing. The detailed procedure and results

are described as follows.

[Methods, Page 36, Line 696]

4.13. Gel permeation chromatography (GPC) analysis

The molecular weight of TPU before and after HME, as well as hot pressing, was analyzed by GPC (Shimadzu Prominence, Kyoto, JPN). TPU samples were dissolved in tetrahydrofuran (THF, ACS reagent grade) at 10 mg/mL final concentration at room temperature under 150 rpm magnetic stirring for 1 h. All TPU samples were filtered using 0.22 μm polytetrafluoroethylene (PTFE) syringe filters and introduced to a GPC equipped with Phenogel 5 μm 10^4 \AA 300 x 7.80 mm GPC/SEC Column (Phenomenex, CA, USA). Ultraviolet and refractive index detectors were used to obtain chromatograms. THF (HPLC grade) was used as an isocratic mobile phase at 1 mL/min flow rate, and the oven temperature was equilibrated at room temperature. Polystyrene was used as a reference standard to calculate molecular weight.

[Results, Page 19, Line 356]

Gel permeation chromatography (GPC) and attenuated total reflectance-Fourier transform infrared spectroscopy (ATR-FTIR) confirmed that the molecular weight and chemical composition of TPU did not change following HME, nor did the additional sample preparation steps for the water contact angle analysis impact polymer composition (Supplementary Fig. 12 & 13).

[Supplementary Information, Page 12, Line 125]

Supplementary Fig. 12. Gel permeation chromatograms (GPC) of TPU before and after hot melt extrusion and hot pressing obtained by using ultraviolet (A) and refractive index (B) detectors. The molecular weight of TPU remained similar post extrusion or pressing. Noise after 10 min elution time determined by refractive index detector are void peaks as different grades of THF (ACS reagent grade and HPLC grade) were used for dissolving TPU.

2) For contact angle determination there was an even further thermal processing stage (hot melt pressing). Again, the surface chemistry may have been modified through thermal degradation/oxidation. It would be good to do GPC analysis as well as ATR-FT-IR and/or NMR analysis of the materials before and post pressing to ensure that the changes observed are not artefacts - even though the control TPU went through the same stages, the presence of spores may have influenced the chemistry of degradation. I note that the UV-Vis spectra are less able to discriminate some of the characteristic degradation outcomes.

Response: We have performed ATR-FTIR analysis of TPU before and after hot melt extrusion, as well as hot pressing and seen no changes in the chemical structure of the TPU. Thank you for bringing this to our attention, as the added data strengthens the manuscript.

[Methods, Page 36, Line 708]

4.14. Attenuated total reflectance -Fourier transform infrared (ATR-FTIR) spectrometry

TPUs were analyzed by ATR-FTIR (Nicolet™ iS50, Thermo Fisher Scientific, Waltham, MA, USA) before and after HME, as well as hot pressing. Spectra was acquired at 4000 - 525 cm⁻¹ with 4 cm⁻¹ spectral resolution at room temperature. 16 co-added scans were averaged. Potassium bromide (KBr) and deuterated-triglycine sulfate (DTGS) were used as beam splitter and detector, respectively.

[Results, Page 19, Line 356]

Gel permeation chromatography (GPC) and attenuated total reflectance-Fourier transform infrared spectroscopy (ATR-FTIR) confirmed that the molecular weight and chemical composition of TPU did not change following HME, nor did the additional sample preparation steps for the water contact angle analysis impact polymer composition (Supplementary Fig. 12 & 13).

[Supplementary Information, Page 13, Line 132]

Supplementary Fig. 13. Attenuated total reflectance-Fourier transform infrared spectroscopy of TPU and BC TPUs before and after hot melt extrusion or hot pressing. BC TPUs contained 0.8 w/w% spores.

From a more general perspective, there were some other issues of note:

1) Throughout the text, the processing should be referred to as mini-twin screw extrusion; it is more like a model of the larger scale process. Likewise, the long (15 min) cycling process adopted is one that is feasible with this apparatus but far less likely in larger scale equipment and then only achieved through a complex screw profile - and that should be noted. Please provide the dimensions of the slit die. Further, polymers are not "incubated" in processing equipment, even those with living organisms incorporated. They are cycled, melt processed, or some other variant of such terms.

Response: We are now using the term "processing" sparingly and replaced it with more specific terminology. We have also clarified that we used a mini-twin screw extruder to fabricate biocomposite TPUs in this work as follows.

[Page 3, Line 50] *High-temperature melt extrusion*

[Page 13, Line 236] *HME using a mini-twin screw extruder (TSE) (Fig. 1)*

[Page 14, Line 257] *above the extrusion temperature*

[Page 14, Line 259] *Full viability recovery after HME*

[Page 15, Line 276] *WT spore cores that were destroyed during extrusion*

[Page 15, Line 277] *HST spores retained their structure following HME*

[Page 28, Line 511] *essentially full viability even after HME*

[Page 35, Line 679] *Shear stress during HME*

We agree with the reviewer, in that the long processing time is not realistic for scale up processes, and also means that the spores will be subjected to less thermal stress upon scale up. We adopted a long processing time (15 min) to ensure the full mixing between the TPU and spores during the melt processing using the bench-top mini-twin screw extruder (**Supplementary Fig. 4**), whose mixing process primarily relies on the cycling of polymer melt through barrel due to the limited screw/barrel length. At the same time, as shown in **Supplementary Fig. 19**, HST spores retained decent viability up to 4-fold increased screw speed (144 rpm) compared to the standard protocol (36 rpm), which depicts that the processing time can be reduced to quarter with marginal risk on the spore viability.

Dimensions of the slit die have been provided. There was marginal expansion or shrinkage of the biocomposite TPU after the extrusion through the slit die.

[Page 13, Line 241] *extrusion through a slit die (slit size: 5.0 mm x 0.7 mm).*

[Page 33, Line 636] *equipped with a slit die (slit size: 5.0 mm x 0.7 mm).*

We have replaced “incubation” with more proper terms for polymer processing throughout the manuscript.

[Page 33, Line 637] *loaded to the TSE and cycled at 135 °C at 36 rpm for 5 min*

[Page 33, Line 639] *The TPU-spore mixture was further processed at 135 °C at 36 rpm*

2) Need to define which standard method or variant of same was adopted for the tensile testing. For example, ASTM D882 is appropriate for sheets/films of <0.7 mm. Was the stretching rate consistent with standard? What load cell was used? How were dogbones cut? (Laser cutting, razor blade with a template - fresh blade every few cuts, other?)

Response: Custom dogbone shape and crosshead rate were used for the tensile testing after rigorously optimizing reproducibility. It is because we used TPU ribbons extruded through 5.0 mm x 0.7 mm slit die as a template for the dogbone cut, and neither ASTM D638, D882 nor D412 was applicable for the TPU samples.

Dogbone specimens were prepared by using a precision knife with a template. The TPU material used in this manuscript (BCF45) was a soft elastomer, and there was little wearing of the razor blade after each specimen cutting. Thus, razor blade replacement did not occur regularly. This information, together with

the load cell information, has been added to the manuscript as follows.

[Methods, Page 34, Line 658] *Biocomposite TPU was tailored into a dogbone shape by using a precision knife with a template for the tensile testing (overall length: > 38 mm; clamping area length: > 10 mm; initial distance between grips: ~18 mm; length of narrow parallel-sided portion: 10 mm; width at ends: 5 mm; width at narrow portion: 2.4 mm; thickness: 0.7 mm).*

[Methods, Page 35, Line 664] *The specimen was stretched at 20 mm/min extension rate until it fractured by using a universal testing machine (Instron 5982, MA, USA) equipped with a 100 N load cell.*

A wide range of crosshead rates has been used from the previous literature to test the tensile properties of TPUs. The extension rate in this work (20 mm/min) was determined by collectively considering the testing time and reproducibility.

3) In a couple of places it is mentioned that samples were collected "from different sites in an exudate". This is unclear. Should give the final dimensions of the ribbon produced post extrusion (there is usually shrinkage plus we don't know how long the extruded ribbon was) and then describe how sites were selected - random along ribbon? middle? edge?

Response: In response to the reviewer's comment, we have newly added a supplementary figure (**Supplementary Fig. 4A**) showing the final shape of the extruded BC TPU and the exemplary sampling sites for the spectrophotometry.

[Supplementary Information, Page 4, Line 73]

Supplementary Fig. 4. (A) Exemplary sampling sites from TPU extrudate for the spectrophotometry (scale bar: 10 mm). UV-Vis spectra of BC TPUs with 0.8 w/w% WT (B) or HST (C) spores and their corresponding controls dissolved/suspended in DMF. Final concentrations of TPU and spore in DMF were 9.92 mg/mL and 0.08 mg/mL, respectively. TPU showed absorbance at ~ 300 nm, while spores absorbed a broad range of UV-Vis after 300 nm. BC TPUs showed characteristic absorbance patterns from both TPU and spore.

4) Should specify the pneumatic pressure applied in hot pressing. I assume the ribbon pieces were used as-is and not placed in any mould or template?

Response: We used 6.3 MPa hydraulic pressure to flatten the samples. As the reviewer mentioned, no mould nor template was used. The pressure information has been added to the manuscript with the corrected pressure source (pneumatic → hydraulic).

[Methods, Page 36, Line 690] A small piece of TPU sample ($\sim 10 \times 5.0 \times 0.7 \text{ mm}^3$) was wrapped in aluminum foil and pressed between two flat heating blocks at $120 \text{ }^\circ\text{C}$ for 10 min through 6.3 MPa hydraulic pressure.

We have also added new photographs of the hot pressing process as **Supplementary Fig. 11A** for the convenience of readers.

[Supplementary Information, Page 11, Line 118]

Supplementary Fig. 11. (A) Sample preparation for water contact angle analysis by using hot pressing (scale bar: 10 mm). Water contact angle of BC TPU^{WT} (B) and BC TPU^{HST} (C) with 0 ~ 1.0 w/w% spore loading. Error bars indicate standard deviations from three independent experiments. (D) Water contact angles were obtained by analyzing the photographs of water droplets on flattened BC TPU^{WT} and BC TPU^{HST}.

5) Should specify source of compost at first use in the experimental. Note what depth compost was sampled from in the commercial facility.

Response: We separated the method for sourcing compost as section 4.17 from the original section (4.20) to define the source of compost at first use in the experimental section.

[Methods, Page 37, Line 731]

4.17. Compost sourcing

Raw compost aged 4–5 months was collected from two industrial composting facilities located in Athens, GA, specifically the University of Georgia and Athens-Clarke County Solid Waste Compost Facility. Landscaping and forest residues, food waste, and livestock manure are inputs for both facilities. The temperature of the composting pile at sampling depth (approximately 30–100 cm) was 46 ± 5 °C. The compost was particle sieved through a 4.76 mm screen. The sieved compost was mixed thoroughly with a resulting pH of 7.4 at the beginning of testing. See Supplementary Table 2 for compost details.

6) How were samples prepared for gravimetric disintegration testing? What dimensions?

Response: The photographs of samples we used for the gravimetric disintegration test are illustrated in **Fig. 5A**. They were $\sim 48.0 \times 5.0 \times 0.7$ mm³ pieces, which corresponded to 200 mg each. We have added the dimensions of the samples to clarify procedural information.

[Methods, Page 38, Line 753] *Initial mass of each TPU piece used for the gravimetric disintegration test was 200 mg with $\sim 48.0 \times 5.0 \times 0.7$ mm³ dimension.*

7) It is unclear to me why you would not do 16s rRNA testing directly on the swabs from the compost or TPU rather than going through a plating process first, given that so many native microbes are not readily culturable and will be missed in this process.

Response: We appreciate your concern regarding the potential missing of native microbes when bypassing the plating process and not directly performing 16S rRNA sequencing on swabs from compost or TPU. The objective of the analysis encompassed not only determining the bacterial population ratios through 16S rRNA sequencing, but also accurately assessing the proportions of strains for the intentionally incorporated strains. This quantification included the characterization of the *Bacillus* sp. among the microbial community and conducting additional marker gene-specific sequencing analysis. Therefore, we initiated the analysis from the plating process; however, consistent with the concerns, we tried to minimize the variability in the initial bacterial community with adequate incubation time and random colony picking.

[Page 22, Line 409] *It should be noted that the microbial analysis was focused on the bacterial*

community^{61,62} considering their abundance in compost. In conjunction with the bacterial consortium analysis, supplementary sequencing analysis of Bacillus sp. was performed to evaluate the presence of the intentionally embedded strains. This process included the initial streaking from a swab onto an LB agar plate and efforts were made to minimize the variability in the initial bacterial community by ensuring ample incubation and employing random colony picking.

Other general comments:

1) There needs to be better justification as to why the biodegradation testing was done at a far lower temperature than typical industrial composting (we are not considering just the maturation phase). This more models home composting, which is fine if made clear. The fact that the Bacillus may not perform as well at the elevated temperatures of commercial composting may be worth noting.

Response: Thanks for the comment. As described in line 377 ~ line 382, plastic recovery is difficult and industrial composting facilities are very scant. Thus, it is highly unlikely that TPU waste would end up in industrial composting, therefore industrial composting conditions were not followed in this work. However, we agree with the reviewer that justifying the temperature condition we used in this work is important, and we have added the following phrases to the manuscript.

[Page 21, Line 382] *In addition, the global infrastructure for industrial composting in a microbe-rich environment is sparse and TPU waste would likely go to environments that are not enriched with TPU degrading microorganisms at the end of its lifecycle. Thus, it is important to test the degradation of TPU in microbially active and, more importantly, less active environments under mild temperature conditions. We simulated these conditions by using microbially active compost and autoclaved compost, respectively, at 37 °C.*

2) The three different levels of magnification in Supp. Fig.2 do not add any additional information - suggest to remove at least one column of images.

Response: We agree with the reviewer. We have decided to show only 10k magnification (middle column).

[Supplementary Information, Page 3, Line 69]

Supplementary Fig. 3. Scanning electron microscope images of lyophilized WT (A) and HST (B) spores before the incorporation into TPU (10k magnification). Each panel represents an individual visualization site. Scale bars are 2 μm .

3) Should be some discussion in the text that even with the data that was used for the Pukanszky model being already trimmed for data below the critical spore concentration, there was still a clear volume percent at which the samples no longer followed the linear relationship: there could be a number of explanations for this, but it should be commented on. It is also worth noting that the Pukanszky model was given for polymer composites containing quasi-spherical particles, which is appropriate for this composite application.

Response: We have discussed the non-linear relationship between the y values of the linear form of the Pukanszky model and volume fraction above certain spore concentrations.

[Results, Page 18, Line 335] *Such decrease in the y value of the linear form of the Pukanszky model above a certain filler content is oftentimes found in composite materials, which is likely due to the increased probability of matrix discontinuity and/or filler agglomeration. Both can behave as defect sites and initiate the failure, negatively affecting the ultimate tensile strength⁵⁵.*

We have also mentioned that the Pukanszky model is appropriate for elucidating the polymer/filler interaction of our BC TPU samples as the spores are quasi-spherical particulates.

[Results, Page 18, Line 320] *The Pukanszky model was given for polymer composites containing quasi-*

spherical particles, which is appropriate for our composite system.

4) There is an odd plateau at around 25 days for cellulose biodegradation in Supplementary figure 11 B). It doesn't seem to match with the results in Supplementary figure 11 A). Is this right? Was there a reason for this?

Response: Thank you for the comment. The plateau is a group of the bottom caps of error bars of several data points around 25 days. We have changed graph formatting to prevent misleading interpretation. At the same time, we extended the respirometry data from 150 days to 180 days.

[Supplementary Information. Page 22, Line 211]

Supplementary Fig. 20. Cumulative CO₂ production (A), absolute biodegradation (B) and relative biodegradation (C) profiles of blank, cellulose, TPU, BC TPU^{WT} and BC TPU^{HST} in compost at 42 °C at 45–60% relative humidity. Error bars indicate the standard deviations from three independent experiments. (D) 180-day averages of cumulative CO₂ production and percent biodegradation values for each test group. Relative biodegradation was calculated from the absolute biodegradation of samples relative to the reference material, cellulose.

Minor notes:

1) Please define the following at first use: GRAS, TSE, LB, PBS, SNP, HME

Response: We have defined the abbreviations at their first uses throughout the manuscript.

2) line 295: it is stress versus strain curves, not vice versa

Response: We have revised it. We have also revised “versus” to “vs.” throughout the manuscript for consistency.

3) lines 179, 258, 307, 391, 395, 416, 442: better to use indicate rather than indicated (the document uses both past and present tense inconsistently at present when indicating outcomes/conclusions). Likewise with "suggested" throughout.

Response: We have overall revised the tense of phrases such as “indicate”, “suggest”, “imply” and “depict” throughout the manuscript for consistency. We are now overall using past tense in the Result section.

4) Line 418: suggest: can successfully germinate with biological activity in compost after fabrication...

Response: We have edited the manuscript as the reviewer suggested [Page 23, Line 430].

5) Line 504: screw geometry generally means screw profile and L/D, not just thickness - in this case you are just talking about barrell and screw diameter. The small-scale conical twin screw in this model system is not meant to be representative of a more complex twin screw profile with back flight, mixing, feeding and other zones. It should be recognised that the actual processing conditions at larger scale will be somewhat different.

Response: We have revised the manuscript and more carefully approach the scale-up potential as follows.

[Discussion, Page 28, Line 516] *The scale-up and scale-down of TSE-based HME is widely studied⁷³, and the throughput of BC TPU fabrication (15 g/h) by the benchtop TSE (Supplementary Fig. 18) can be potentially increased to larger scales. For example, our data showed that spores retain significant survivability (~88 %) up to 144 rpm screw speed, which corresponds to up to 220 MPa shear stress (Supplementary Fig. 19). Increasing the screw speed, together with an increase in the screw and barrel size will achieve exponential upscaling. Under volumetric scaling, the throughput can be increased to the power of 3 with an increase in outer screw diameter (D)^{74,75}, yet the various relevant processing*

parameters are needed to be considered and optimized for the successful scale-up^{76,77}.

6) line 559: introduced into the HST

Response: We have revised the typos throughout the manuscript including the reviewer's suggestion. We have noted the current line numbers for each modification for the reviewer's reference.

[Line 579]

7) line 593: 0.025 L

[Line 613]

8) line 676: the polymer matrix dissolved, not incubated, in 10 mL DMF

[Line 717]

9) line 679: spore suspension was further soaked at room temperature

[Line 719]

10) line 694: followed by autoclaving

[Line 743]

11) line 696: after autoclaving

[Line 745]

12) line 707: the sieved compost was mixed thoroughly

[Line 737]

Supplementary:

page 11: legend: with 45-55% relative humidity.

[Supplementary Information, Page 16, Line 163]

page 11: legend: spore loading. (B) Mass loss profile. Error bars indicate....

[Supplementary Information, Page 16, Line 160]

page 13: watch the significant figures in supplementary figure 11 D). In addition, line 156: specify that

this is biodegradation relative to cellulose. Note: it is unclear if the shaded areas on the nonlinear curves represent the 95% confidence intervals for each sample or just shading between the three individual curves.

Response: We have revised the significant figures in the **Supplementary Fig. 20D**. We have also described how relative biodegradation is calculated. Shaded area is a group of overlapped error bars. We have revised the graph formatting.

[Supplementary Information, Page 22, Line 211]

Line 162, starting with "As a result" is odd, and doesn't really follow from the previous text.

Response: We have revised the sentence [Supplementary Information, Page 23, Line 220].

Supplementary figure 13 is not really clear. There are a lot of samples represented by different shapes (triangles, circles, diamonds etc.) without any definition as to what these symbols represent - we only know what the colour family indicates. Were other commercially available TPUs outside the Elastollan range included? If so, where did the commercial data come from?

Response: **Supplementary Fig. 17** has been revised with more detailed symbol definition and corresponding references.

[Supplementary Information, Page 19, Line 193]

Supplementary Fig. 17. Elongation at break and tensile stress of commercially-available TPUs (grayscale) were plotted with BC TPUs (colored) developed in this work. Previously reported biodegradable TPUs are plotted with green symbols³⁻¹³. Tensile properties of commercially-available TPUs were adapted from the BASF Elastollan[®] product range. The trade-off barrier (dashed line) was plotted by using the tensile properties of top 15 commercial TPUs that mutually exhibit high tensile stress and elongation at break.

Finally, in supplementary table 2, the solids content for the compost for respirometry seems low. Was this a wetter compost? If so, why?

Response: There was an error in the calculations for reported volatile solid, total solid, and ash content. We have corrected the error and **Supplementary Table 2** now reflects the updated values for both compost inoculates. The compost used in the respirometry tests retained higher moisture due to rain events prior to sourcing.

Reviewer #2 (Remarks to the Author):

In this manuscript, Kim et al. generated a new kind of engineered living material (ELM), which was composed of evolved *Bacillus subtilis* spores and thermoplastic polyurethane (TPU). This biocomposite could be characterized and functionalized in different ways. First, the authors carried out adaptive laboratory evolution of *B. subtilis* ATCC 6633 spores to successfully improve the property of heat-shock tolerance (up to 135°C during hot melt extrusion). Next, the authors found two of the commonly mutated genes (*fusA* and *abrB*) that would enhance the heat-shock tolerance. Then, the fabrication of biocomposite by using TPU and evolved spores (spores of evolved A5_F40_I1 strain) was successfully performed and the tensile properties were evaluated in four different ways. Finally, such biocomposite TPU could be utilized to facilitate disintegration and genetically express GFP within cells. In conclusion, this manuscript exhibited a composite material consisted of living spores and nonliving TPU, with compelling mechanical properties and expanded biological applications.

Overall, the experiments in this work are well designed and performed, the data are solid and convincing, and the manuscript is well-written and clearly presented. Specially, the newly produced ELM by *B. subtilis* spores were embedded into polymers, rather than biofilms (Huang et al. Nature Chemical Biology, 15, 34-41, 2019; Zhang et al. Materials Today, 28, 40-48, 2019), agarose (González et al. Nature Chemical Biology, 16, 126-133, 2020), or biomineralized scaffolds (Kang et al. Nature Communications, 12, 7133, 2021), which provided another approach to construct *B. subtilis* spores-based ELMs. Such design may become a demo and provide inspirations for the further researches in this field. **Taken together, I highly recommend this manuscript for publication in Nature Communications after addressing the following issues.**

Major concerns:

1. Page 4, line 75: The statement of “Despite this potential, live cells have rarely been exploited as polymer additives in practice due to their fragility.” is not appropriate. In recent years, there were many polymer-based ELMs that have been reported (e.g., bacteria in calcium alginate gel: Chen et al. Nature Chemical Biology, 18, 289-294, 2022; Peng et al. Advanced Materials, 2305583, 2023). I suggest the authors to revise this sentence.

Response: We have revised our manuscript after carefully reviewing the references the reviewer provided.

[Page 4, Line 75] ~~Despite this potential, live cells have rarely been exploited as polymer additives in practice due to their fragility.~~ However, live cells are fragile and require careful handling in terms of hydration, osmotic pressure, temperature and pH, compared to other non-living biological substances.

2. Page 4, line 82: The statement of “and in most studies it is unclear if the cells maintained viability.” is not appropriate. As far as I know, most of published papers about ELMs contain the experimental results of cell viability (e.g., Tang et al. Nature Chemical Biology, 17, 724-731, 2021). I suggest the authors to

revise this sentence.

Response: Thank you very much for the information about their work. We have removed the sentence.

3. Page 5, line 94: In my opinion, the statement here of probiotic properties on *Bacillus subtilis* is not suitable, because this manuscript lacked the utilization of its probiotic properties. I recommended the authors to replace the properties of “probiotic” with “GRAS (page 25, line 497)” here, and move the statements of probiotic properties to the Discussion section.

Response: We agree with the reviewer. We have revised the manuscript accordingly.

[Introduction, Page 5, Line 93] *They are ubiquitous in nature and FDA-approved generally recognized as safe (GRAS) substances^{23,24}.*

[Discussion, Page 28, Line 502] *It should be also noted that B. subtilis is not only beneficial for human health as probiotics⁶⁸⁻⁷⁰, but also aids the growth of plants as biocontrol agents⁷¹.*

4. In Figure 3A, the photographs lacked scale bars. I suggest the authors to revise and add scale bars because it is important to indicate the lengths and diameters of the BC TPU fibers.

Response: We have added the scale bars to **Fig. 3A**. We have also added the image of the extruded BC TPU in **Supplementary Fig. 2** with the final dimension.

[Page 16, Line 279]

Fig. 3. (A) Photographs of BC TPU^{WT} (inset) and BC TPU^{HST} with 0, 0.2, 0.4, 0.6, 0.8 and 1.0 w/w% spore loadings, left to right (scale bars: 10 mm).

[Supplementary Information, Page 2, Line 63]

Supplementary Fig. 2. Full scale image of extruded BC TPU^{HST} with 0.8 w/w% spore loading (scale bar: 10 mm). The width and thickness of the extrudate was determined by the geometry of the slit die (5.0 mm)

x 0.7 mm). The final length of extrudate depended on the extrusion time and generally resulted in ~750 mm.

5. In the body text of this manuscript, the “Supplementary Fig. 11” was lacked between “Supplementary Fig. 10” and “Supplementary Fig. 12”. I suggest the authors to rearrange the order of Supplementary Figures in the revised manuscript files.

Response: We have relocated **Supplementary Fig. 11** to **Supplementary Fig. 20**, and cited in the manuscript at Page 29, Line 537. We have overall checked that all the tables and figures are mentioned in the body text of the manuscript.

Minor concerns:

1. In Fig. 1, the “Hot-melt Extrusion” included the symbol of “-”, however, the other statements in this manuscript were the same as “hot melt extrusion”. Please unify the format.

Response: We have overall revised the manuscript for consistency.

2. In Supplementary Information, line 2 of “Table of Contents”, there needs a space between “ATCC” and “6633”.

Response: We have overall revised the manuscript for consistency.

3. Page 11, line 222: When the statements of “A3_F40_I1” was firstly presented in this manuscript, it needs a detailed explanation, like “A3: ALE lineage 3; F40: Flask 40; I1: Isolate 1”. Such explanation will help readers to clearly understand the meanings.

Response: We have added the detailed explanation.

[Page 12, Line 219]

The isolate A3_F40_I1 (A3: ALE replicate #3, F40: Flask #40, I1: Isolate #1),

4. Page 12, line 243: The statement of “twin-screw extruder” included the symbol of “-”, however, the other statements in this manuscript were the same as “twin screw extruder”. Please unify the format.

Response: We have revised this throughout the manuscript.

5. Page 13, line 268: The typo of “using a X-ray microscope” should be corrected as “using an X-ray microscope”.

Response: We have corrected the typo.

6. Page 16, line 320 and 327: Please carefully correct the authors’ names as “Pukánszky”, rather than “Pukanszky”.

Response: Thanks for the comment. We have carefully reviewed and corrected the Author’s name (Pukánszky) throughout the manuscript and supplementary information.

7. Page 24, line 464: The statement of “heat shock” lacked the symbol of “-”, however, the other statements in this manuscript were the same as “heat-shock”, including a symbol of “-”. Please unify the format.

Response: We have overall revised the manuscript for consistency.

8. Page 25, line 483: The abbreviation of “non-GMO” should be clarified here for readers, may be “non-genetically modified organism”.

Response: We have defined the abbreviations at their first uses throughout the manuscript, such as GMO, GRAS, TSE, etc.

9. Page 25, line 497: The abbreviation of “GRAS” should be clarified here for readers, may be “Generally Recognized as Safe”.

Response: We have clarified the full name of GRAS at its first use (Page 5, Line 93).

10. Page 25, line 500: The abbreviation of “TSE” should be clarified here (Twin Screw Extruder?), or move to the above of this manuscript when the full name was firstly presented.

11. Page 27, line 533: The typo of “have not surpassed the trade-off-barrier” should be corrected as “have not surpassed the trade-off-barrier”.

Response: We have clarified the full name of TSE and corrected the typo.

Reviewer #3 (Remarks to the Author):

The manuscript by Han Sol Kim et al. presents a novel method for fabricating Bacillus spore-filled thermoplastic polyurethane (TPU) biocomposite. Temperature resistance of Bacillus subtilis spores was enhanced through Adaptive Laboratory Evolution (ALE) to fabricate novel ELMs via hot melt extrusion and to reinforce the TPU matrix's mechanical properties.

Developing a novel strategy to incorporate living cells into thermoplastic material could allow the utilization of ELMs for various applications and advance the additive manufacturing of ELMs.

Although the topic of this study is important, this manuscript has overinterpreted the presented data and possesses limited novelty. The authors have provided a lot of data from tedious experiments, but unfortunately, it does not clearly support the hypothesis and does not make this manuscript better.

Due to the many concerns, I have stated below, I would not recommend the publication of this manuscript in Nature Communication.

Major concerns:

1. Tensile properties of biocomposite TPUs.

First, the particular tensile properties of TPU WT and HST should be incorporated together on one graph. It is difficult to compare them if they are not on the same graph.

Second, in line 306, the authors claim that the results (difference between WT and HST) are remarkable. I could not call the 12% difference in toughness remarkable.

From the graphs, it looks like WT TPU has lower spores (0.4%), resulting in higher toughness than 0.4% of HST TPU. You need to add double the amount of HST TPU spores (0.8%) to achieve slightly better toughness. This needs to be addressed in the manuscript, and claiming that the mechanical properties of HST TPU are remarkable is an overstatement.

Third, the HST TPU shows only a 10% increase in tensile test and a 1% increase in elongation break. These results also are not exciting to claim such remarkable mechanical properties improvement.

Response: Thank you for your comments. This is the first report demonstrating that spore incorporation can enhance the mechanical properties of TPU. Thus, our main focus is comparing the TPU without spore versus spore bearing biocomposite TPUs. We attached a graph that combined the toughness of biocomposite TPUs with WT and HST spores and their corresponding controls (TPUs without spores) for the reviewer's reference. BC TPU^{WT} and BC TPU^{HST} have their respective baseline TPUs, which were prepared from the same batches of BC TPU^{WT} or BC TPU^{HST} fabrication. This information has been added to the method section. There was no statistical significance between the toughness of baseline TPUs for BC TPU^{WT} and BC TPU^{HST}. It should be also noted that the relative toughness improvement of TPU

material by the addition of WT spores was lower than that of HST spores because WT spores are mostly denatured post HME.

[Supplementary Information, Page 8, Line 102]

Supplementary Fig. 8. Toughness of biocomposite TPUs with WT and HST spores and their corresponding controls combined in one graph.

[Methods, Page 35, Line 674] *Tensile properties of BC TPU^{WT} or BC TPU^{HST} were compared with their respective baseline TPUs, which were prepared on the same day of BC TPU^{WT} or BC TPU^{HST} fabrication via HME, respectively.*

We agree with the reviewer that some readers might not consider the 12% toughness difference between BC TPU^{WT} and BC TPU^{HST} to be remarkable. We have revised the sentence as follows.

[Page 17, Line 305] *,while the ultimate toughness improvement of BC TPU^{HST} (37%) also outperformed that of BC TPU^{WT} (25%) at their respective critical spore concentrations.*

Again, we believe that TPU is a more appropriate benchmark for BC TPU^{HST}, rather than BC TPU^{WT}. For the most part the wild type spores have low viability and are not a viable candidate for composite formation because of this. As described in the Discussion (Line 538), the baseline TPU materials used in this work (BCF 45) is a commercial grade TPU manufactured by BASF, which showed excellent tensile properties among TPU materials in the market. The incorporation of spores enhanced the tensile properties of BCF45 beyond the trade-off barrier between the tensile strength vs. elongation at break (**Supplementary Fig. 17**), and it can be a meaningful result.

2. Facilitated disintegration of biocomposite.

Here, the authors wrote that at the end of the life cycle of a BS TPU, spores can be germinated to facilitate TPU disintegration.

First, the authors did not provide what % of spores integrated into TPU were carried in described degradation experiments.

Second, the difference in mass loss between WT and HST is 29.1% after 5 months in autoclaved compost. Why there is so little difference?? This difference should be significantly larger if WT had only 20% viable cells and HST ~100%. I assume the HST variant has negatively altered metabolic activity, so they cannot degrade TPU efficiently.

Third, in Supplementary Fig 10B. authors show that TPU alone, TPU-WT, and TPU-HST show the same level of mass loss after 5 months in non-autoclaved compost. No difference. It means that even TPU alone, wherein spores are absent, degrades to the same extent. Thus, the statement that engineered material facilitated biocomposite's disintegration seems flawed. The results do not support the hypothesis.

Response: We have added the spore loading of BC TPU samples we used for the disintegration test in the Method section. Thanks for the comment.

[Methods, Page 38, Line 754] *At least 30 pieces of each TPU or BC TPU sample were incubated in the untreated and autoclaved compost. Spore loading of BC TPU^{WT} and BC TPU^{HST} samples used for the disintegration test was 0.8 w/w%.*

On the second point, it is difficult to stoichiometrically correlate the initial CFU with their biological activity given the nature of growth of microbes being anywhere from linear to exponential and the compost environment being highly variable. However, we have presented additional experiments in **Supplementary Fig. 6**, which confirm that there are no major phenotypic differences in measurable metabolic activity related to cell growth, TPU assimilation and spore viability after evolutionary engineering of ATCC 6633 WT when culturing in a controlled environment where such comparisons are possible.

[Results, Page 14, Line 260] *Notably, this enhanced heat tolerance was achieved without observation of phenotypic tradeoffs when compared to the parental strain, as both WT and HST strains exhibited similar cell growth profiles under multiple liquid media environments including TPU assimilation (Supplementary Fig. 6).*

[Supplementary Information, Page 6, Line 87]

Supplementary Fig. 6. Cell growth profile of ATCC 6633 WT and HST strains under M9 minimal medium supplemented with general carbon sources such as 4 g/L glucose (M9 + Glucose) or 4 g/L glycerol (M9 + Glycerol) (A) and complex media such as LB or DSM (B) at 37 °C under 250 rpm shaking. (C) TPU assimilation activity determined by cell growth under M9 minimal medium supplemented with 10 g/L TPU powder as the sole nutrient source at 37 °C under 250 rpm shaking. (D) Spore viability of ATCC 6633 WT and HST strains.

For the little degradation rate differences among samples in untreated compost, non-autoclaved compost simulates the degradation substrate with high degradation activity, but plastic wastes cannot always be collected and processed under such ideal degradation conditions. The majority (> 80%) of plastic waste is escaping our collection/recycling effort (Page 21, Line 337), and they would likely go to environments that are not enriched with polymer degraders. In this work, we incorporated TPU-degrading bacteria directly into the TPU matrix and showed that BC TPU^{HST} was degraded fast (> 90% after 5 months) regardless of the microbial activity of the incubating substrate (i.e. autoclaved compost and non-

autoclaved compost). The disintegration rate of BC TPU^{HST} was ~2-fold faster than the baseline materials (TPU). This result is meaningful because the accumulation of orphan TPUs in nature can be potentially mitigated as the embedded TPU degraders (i.e. *B. subtilis*) can trigger the disintegration of TPU waste stimulated by the moisture and nutrients abundant in nature.

3. In line 117 – The authors say they are “programming/facilitating degradation of a spore-filled biocomposite TPU”. This claim is overstated. The degradation is only facilitated, *Bacillus* has natural degradation activity against polyester. In this work, the degradability of TPU is not programmed or engineered but rather evolved.

Response: We believe that we programmed the degradation of BC TPU by utilizing the stimuli-responsive germination behavior of spores. Uncontrolled degradation of polymers can negatively affect the lifetime and reliability of plastic products. Spores are metabolically dormant and do not mediate the degradation of TPU until they are exposed to the nutrients and moisture and germinate. Thus, we could program the degradation of biocomposite TPU to its end of life in the soil environment.

4. In line 135 – “Overall, this work presents a scalable method for fabricating biocomposite materials with improved mechanical properties and programmed biological functionalities.” There are no programmed biological functionalities. GFP fluorescence in *Bacillus* pores has already been shown in González, L.M., Mukhitov, N. & Voigt, C.A. Resilient living materials built by printing bacterial spores. *Nat Chem Biol* 16, 126–133 (2020)

Response: We agree with the reviewer that we have used “program” interchangeably throughout the manuscript and it can confuse readers. Spore forming bacteria can naturally program their metabolic activity based on the environmental stimuli. On the other hand, foreign biofunction(s) can also be potentially programmed to live bacteria via genetic engineering. To clearly distinguish these two concepts, we have used either the expression “genetically program”, “rational program” or “external biofunction” for the latter case.

[Page 3, Line 55] [Page 6, Line 129] [Page 25, Line 443] [Page 25, 444]

5. The authors use ALE experiments to increase the heat tolerance of *B. subtilis* spores. The experiment was successful, giving survivability of about 96-100% compared to 20% WT spores after hot melt extrusion at 135 C. Next, the authors focused on identifying the mutations that lead to heat tolerance. I find this study interesting. However, the authors missed the most important point that it is the most significant to develop novel ELMs.

To develop functional ELMs, living bacteria should be viable and metabolically active to perform

programmable functions. The ALE, as the evolutionary method, can influence metabolic activity significantly. The authors focused only on variants with growth rates and viability similar to WT. Still, they did not test if they alter metabolic activity, which is very easy to check via various assays. I find it a large overlook that can lead to developing ELMs that could not be functional even if bacteria are alive.

Response: We disagree with the reviewer that we ‘missed the most important point’. ELMs are an emerging field and as such, are defined in a variety of ways. It is very clear that the bacteria are stabilized in their spore form and germinate to propagate and produce a variety of functions upon introduction of nutrients. Aiming to use ELMs to solve the plastics pollution problem is a noble effort in our opinion and we hope that the reviewer can have a more open mind as to the definition of an ELM given the importance of the problem. We believe that it will make the field of ELM more diverse and interesting.

To answer the more specific points regarding metabolic activity, the evolved strains were directly shown to be robust. Specifically:

1. The evolved strains were grown repeatedly from a frozen stock from a low inoculation density to a high density several times. This data was presented in **Supplementary Fig. 6** and **Supplementary Fig. 14**. Thus, the evolved cells were clearly metabolically active, converting substrates in the media by catabolically transforming them to biomass and energy in every condition relevant to the project and application.
2. Furthermore, see the comment below that the evolved cells did not display a tradeoff in gaining heat shock tolerance vs. being metabolically active that can be inferred from them having a similar cell growth profile under various media, TPU assimilation, and spore viability. These are strong indicators of metabolic activity.

6. Line 264 – The authors wrote that WT and HST strains exhibited similar specific growth rates, yields, and cell viability. No data is provided. It should be provided as a supplementary figure.

Response: We have added a new supplementary figure (**Supplementary Fig. 6**) that shows the fermentation profiles in minimal media with general carbon sources (glucose and glycerol) and complex media. Additionally, we have presented the fermentation profile in minimal media with TPU as a sole carbon source and spore viability to ensure that there was no discernible tradeoff in evolved strains.

[Supplementary Information, Page 6, Line 87]

7. Figure 3D-F – The TPU and BC TPU images are of odd, slopy shapes. The authors did not care to fabricate a composite with a uniform shape. I personally believe that better samples and images are suitable for publication in journals like Nature Communication.

Response: A uniform tape was fabricated through twin screw extrusion (**Supplementary Fig. 2**), as would be seen in the polymer processing industry. For the specific panels chosen by the reviewer, the X-ray microscope we used (Xradia 510 Versa) requires a small bar-shape sample with $\sim 1 \times 1 \text{ mm}^2$ cross-section and $\sim 10 \text{ mm}$ length, and we prepared the samples by cutting the TPU and BC TPU extrudates using a razor blade. We have added this information to the Method section.

[Methods, Page 34, Line 650] *Samples were tailored into $\sim 1.0 \times 1.0 \times 10 \text{ mm}^3$ bar shapes using a razor blade to be fitted into the sample holder of XRM.*

8. Programming biofunction.

Authors genetically engineer HST strain to produce GFP. It has been done previously at González, L.M., Mukhitov, N. & Voigt, C.A. Resilient living materials built by printing bacterial spores. *Nat Chem Biol* 16, 126–133 (2020). The authors provide fluorescence images without fluorescence quantification. It is difficult to spot differences between biocomposite in LB and compost. The material's functionality is shown only at the end of ELMs' life after adding it to compost. The point of developing ELMs is to show functional material that can perform, not perform only during degradation.

Response: Again, we ask the reviewer to have an open mind about the potential incarnations of ELMs. The biological components of BC TPU could sense the environmental changes (moisture and nutrients), and enhanced the degradation of TPU to mitigate plastic accumulation. Such stimuli-responsiveness (germination) and beneficial biofunctionality (TPU disintegration) align well with the mission of developing living materials. In other words, the ELM developed in this manuscript performs after the lifetime of the plastic, while hibernating during the lifetime of TPU.

CLSM analysis in this manuscript is qualitative because we focused on validating if the rationally-programmed *B. subtilis* can express the plasmid after biocomposite fabrication. We have mentioned in the main text that the CLSM is qualitative. The point here was to show that an introduced gene could survive the stressors of the extrusion process. While we understand that GFP does not perform much in the way of a useful function, it does allow for a clear visualization of protein expression. And we agree with the reviewer that it is difficult to spot a difference between the LB and compost, this is the point. The gene can be expressed in a non-optimized environment in compost extract rather than LB. In the future, genes with more apparent function will be incorporated to enhance biodegradation.

[Results, Page 25, Line 452] *CLSM imaging qualitatively revealed the absence of green fluorescence signal from the TPU and BC TPU^{HST} incubated in PBS or LB (Supplementary Fig. 16).*

Minor concerns:

1. In the introduction, the authors claim that ELMs are composed of living cells combined with composite

materials. It is not quite correct. Authors describe hybrid ELMs. Besides hybrid ELMs, “autonomous” ELMs are made from engineered cells to produce functional material that embeds living cells. Autonomous ELMs started and pioneered this field. The authors do not mention it and do not cite any relevant literature.

Response:

We have defined the composite materials composed of live cells and synthetic materials as hybrid engineered living materials.

[Page 3, Line 44] *The field of hybrid engineered living materials (ELMs) seeks to pair living organisms with synthetic materials to generate biocomposite materials with augmented function since living systems can provide highly-programmable and complex behavior.*

[Page 4, Line 60] *Hybrid engineered living materials (ELMs) is a burgeoning field in which living and synthetic matter are combined to provide composite materials with augmented and complex functions, far beyond what a traditional polymeric material could accomplish alone.*

We have added a reference to cover the emerging autonomous ELMs in the introduction.

[Page 4, Line 71] *Moreover, cells can be genetically programmed to autonomously synthesize both small and large molecules and can be further engineered to render other diverse functionalities^{11,12}.*

2. Supplementary Figure 2. The SEM images of spores show lyophilized images of spores. The manuscript suggests that they are images of spores after fabrication with TPU. It is misleading. It could be useful to show SEM images of spores incorporated with TPU and a cross-section of this hybrid material.

Response: We have relocated the citation of Supplementary Fig. 2. (current Supplementary Fig. 3) in the main manuscript file to prevent the mislead. We have also described in the figure caption that Supplementary Fig. 3 represents SEM images of spores before being incorporated into TPU.

[Page 13, Line 243]

Lyophilized spores possess an oblong shape with a ~500 nm diameter and ~1 μm length (Supplementary Fig. 3) and, thus, can serve as submicron particulate fillers if spores remain intact during the extrusion process (Supplementary Fig. 2).

[Supplementary Information, Page 3, Line 69]

Supplementary Fig. 3. Scanning electron microscope images of lyophilized WT (A) and HST (B) spores

before the incorporation into TPU (10k magnification). Each panel represents a different visualization site. Scale bars are 2 μm .

We have tried several methods (cryo-tomog and cryo-fracturing) to visualize the cross-section of BC TPU, but it was difficult to identify spores in BC TPU via SEM due to the low spore amount relative to TPU material (up to 1 w/w%). X-ray microscopy can be a good alternative to visualize spores embedded in TPU as represented in Fig. 3D-F and Supplementary Fig. 7.

3. Figure 3B, and C. should show the viability of WT and HST spores after HME. Graphs show the highest viability of cells with 0% of spore content. I believe it is a mistake, and this is a control - spore viability before HME treatment.

Response: We have revised Figure 3B and C. Thanks for catching our mistake. The controls are now labeled as WT spores* or HST spores*. Asterisk represents that the WT and HST spore controls followed the same solvent treatment procedures with the spores in BC TPUs. This information is provided in the figure caption.

[Page 16, Line 279]

Fig. 3. (A) Photographs of BC TPU^{WT} (inset) and BC TPU^{HST} with 0, 0.2, 0.4, 0.6, 0.8 and 1.0 w/w% spore loadings, left to right (scale bars: 10 mm). The viability of WT (B) and HST (C) spores before and after HME. Spores were extracted from BC TPUs by dissolving the TPU compartment using DMF (no

spore loss due to extraction assumed). Asterisk represents that the WT and HST spore controls followed the same solvent treatment procedures with the spores in BC TPUs. Error bars indicate the standard deviations from three independent experiments. (D-F) MicroCT (XY projection) images of TPU and BC TPUs with 0.8 w/w% WT or HST spores (scale bars: 500 μm).

REVIEWERS' COMMENTS

Reviewer #1 (Remarks to the Author):

Having reviewed the updated manuscript and the response to reviewers I am satisfied that the authors have in the main adequately addressed the concerns raised and improved the manuscript sufficiently to be worthy of publication.

My one concern is that Reviewer #3 was right to be critical of some of the claims, particularly around "programming biodegradation". To some extent this is semantics, but reviewer #3 does have a point in that biodegradation is only initiated under the right conditions (the author's point) but it is not really "programmed" biodegradation, which is biodegradation that is timed to be rapid and specific and can be dialled up rather than be condition dependent. Instead it is more biodegradation that is facilitated by the germination of the spores under the right environmental conditions. This is still uncontrolled biodegradation, just facilitated. I suggest the authors tighten up this text.

Reviewer #2 (Remarks to the Author):

The authors have carefully and correctly addressed all my concerns, so I recommend this manuscript for publication on Nature Communications without further modifications.

Reviewer #3 (Remarks to the Author):

Thank you for submitting the revised version of the manuscript and detailed responses to my comments. My concerns are addressed by the authors in the revised manuscript.

Please see our point-by-point response to the reviewers' comments. Comments from the reviewers are in regular font, our response is in blue, and text in red indicates changes made to the text.

Reviewer(s)' Comments to Author:

Reviewer #1 (Remarks to the Author):

My one concern is that Reviewer #3 was right to be critical of some of the claims, particularly around "programming biodegradation". To some extent this is semantics, but reviewer #3 does have a point in that biodegradation is only initiated under the right conditions (the author's point) but it is not really "programmed" biodegradation, which is biodegradation that is timed to be rapid and specific and can be dialled up rather than be condition dependent. Instead it is more biodegradation that is facilitated by the germination of the spores under the right environmental conditions. This is still uncontrolled biodegradation, just facilitated. I suggest the authors tighten up this text.

Response: Thank you for your comment. In response, we have revised our manuscript as follows.

[Introduction, Page 4]

Collectively, the high stability, safety, polyester degradation activity, and triggerable sporulation/germination of *B. subtilis* make them promising additives in developing biocomposite polymers with programmed biodegradation.

→ Collectively, the high stability, safety, polyester degradation activity, and triggerable sporulation/germination of *B. subtilis* make them promising additives in developing biocomposite polymers with **facilitated** biodegradation.

Reviewer #2 (Remarks to the Author):

The authors have carefully and correctly addressed all my concerns, so I recommend this manuscript for publication on Nature Communications without further modifications.

Response: Thank you for reviewing our manuscript. Your comments helped us to improve our manuscript.

Reviewer #3 (Remarks to the Author):

Thank you for submitting the revised version of the manuscript and detailed responses to my comments. My concerns are addressed by the authors in the revised manuscript.

Response: We appreciate the reviewer for carefully reviewing our manuscript.